# Robust Parallel Diffusion Sampling via Dynamic Jacobian Bandwidth

**Zile Huang** [1]   **Ser-nam Lim** [1]

## Abstract

Recent years have witnessed significant progress in developing effective diffusion models. Parallel sampling is a promising recent approach that reformulates the sequential denoising process as solving a system of nonlinear equations, and it can be combined with other acceleration techniques. However, current progress is limited by the trade-off between high fidelity and computational efficiency. This paper addresses the challenge of scaling to high-dimensional, multi-modal generation. Specifically, we present ROPA (Robust Parallel Diffusion Sampling), which takes into account the properties of the denoising process and solves the linear system using adaptive local sparsity to achieve stable parallel sampling. Extensive experiments demonstrate ROPA's effectiveness: it significantly accelerates sampling across diverse image and video diffusion models, achieving up to $2.9\times$ speedup with eight cores, an improvement of 20.8% over baselines without sacrificing sample quality. ROPA enables parallel sampling methods to provide a solid foundation for real-time, high-fidelity diffusion generation.

## 1. Introduction

Over the past few years, the landscape of generative modeling has been reshaped by the ascent of Diffusion Models (Ho et al., 2020; Song et al., 2020b). These models have emerged as a pivotal methodology for diverse applications (Chung et al., 2023; Yang et al., 2024a; Esser et al., 2024; Ma et al., 2024; Polyak et al., 2025), spanning from high-quality image/video generation to molecular generation. Despite remarkable success, diffusion models require many sequential denoising steps for generating high-quality samples, each involving expensive neural network evalua-

tions. This sequential dependency severely limits inference speed, particularly for real-time applications and large-scale deployment scenarios. Previous works have explored faster numerical Stochastic differential equations (SDEs) or Ordinary differential equations (ODEs) solvers like DDIM (Song et al., 2020a) and DPMsolver (Lu et al., 2022), distilling the ODE trajectory into neural networks (Salimans & Ho, 2022) or straightens trajectories via Rectified Flow (Lipman et al., 2023). Others develop sparse-attention and attention cache (Zhang et al., 2025; Zou et al., 2025).

**Diffusion Models** are generative models built on a foundation of two processes: a forward process that systematically corrupts data into noise, and a reverse process that learns to reverse this corruption to generate new data. This dynamic is elegantly described by SDEs. Considering a clean image $\boldsymbol{x}_0$ sampled from the real data distribution, the forward process gradually perturbs this image with noise over a continuous time interval $t \in [0, T]$, transforming it into a sample $x_t$ that follows a simple prior distribution, like a standard Gaussian. This noising process is defined by the following SDE:

$$\mathrm{d}x_t = f(t)x_t \,\mathrm{d}t + g(t) \,\mathrm{d}w, \qquad (1)$$

where $\mathrm{d}w$ indicates the standard Wiener process. Although the formulation is expressed in continuous time, in practice we are solving a discrete nonlinear system due to the numerical discretization of the SDE. Then, to generate the corresponding clean latent from the easily sampled random noise, we have to reverse the forward SDE in Eq. 1, resulting in the following reverse SDE formulations:

$$\mathrm{d}x_t = \underbrace{\left[f(t)x_t - g^2(t)\nabla_{x_t} \log p(x_t)\right]}_{\varphi(x_t,t)} \mathrm{d}t + \underbrace{g(t)}_{\sigma_t} \mathrm{d}w, \quad (2)$$

where $\nabla_{x_t} \log p(x_t)$ can be approximated by a score function $S_\theta(\cdot)$, parameterized by a neural network with learnable weights of $\theta$; $\varphi(x_t, t)$ denotes the drift function for the reverse diffusion process; $\sigma_t$ represents the corresponding coefficient of diffusion counterpart. Let $\Phi(t, s, x_s)$ represent an integral result of $x_t$ by Eq. 2 over a time interval from $s$ to $t$, with an initial value $x_s$:

$$\Phi(t, s, x_s) = x_s + \int_s^t \varphi(x_\tau, \tau) \,\mathrm{d}\tau + \int_s^t \sigma_\tau \,\mathrm{d}w. \quad (3)$$

[1]University of Central Florida, Orlando, United States. Correspondence to: Ser-nam Lim <sernam@ucf.edu>.

*Proceedings of the 43$^{rd}$ International Conference on Machine Learning*, Seoul, South Korea. PMLR 306, 2026. Copyright 2026 by the author(s).

Consequently, the analytical solution of Eq. 2 at time $t$ can be expressed as

$$x_t = \Phi(t, 0, x_0), \quad x_0 \sim \mathcal{N}(0, I), \tag{4}$$

where $\mathcal{N}(0, I)$ denotes the standard Gaussian distribution.

**Formulating Diffusion Sampling to Solving Non-linear Equation.** Recent advances in parallel sampling (Shih et al., 2024; Tang et al., 2024a; Lu et al., 2025) have shown promise by reformulating the sequential process as solving systems of nonlinear equations, enabling simultaneous computation across multiple timesteps. Existing parallel sampling algorithms establish the following system of non-linear equations to reformulate the integral-based formulation of the diffusion model on a discrete grid $\{t_0, \ldots, t_T\}$:

$$x_{t_{n-1}} - \mathcal{F}_n^{(w_n)}(x_{t_n}, \ldots, x_{t_{n+w_n-1}}) = 0, \tag{5}$$

where $w_n$ is the window size (number of future states coupled) at step $n$. $\mathcal{F}_t^{(i)}$ denotes a solver for estimating results in timestamp $t$ with acknowledging previous states, i.e., $x_t, \cdots, x_{t-i}$. The sampling methods utilize an iterative refinement manner to gradually adjust an estimation trajectory $\{\hat{x}_t, t = 0, \cdots, T\}$. Each state from the trajectory $\{x_t, t = 0, \cdots, T\}$ is first initialized with noise value, denoted as $\left\{\hat{x}_t^{(0)}, t = 0, \cdots, T\right\}$. Denote by $\hat{x}_t$ the vector, $\hat{x}_{0:T} = [\hat{x}_0^\top, \cdots, \hat{x}_T^\top]^\top$. Then, for the $k^{th}$ parallel iteration, where integer $k \in [0, K]$, a Newton-type update refines the variables as

$$\hat{x}_{0:T}^{(k+1)} = \hat{x}_{0:T}^{(k)} - \mathcal{G}^{(k)} \mathcal{R}_{0:T}^{(k)}, \tag{6}$$

where $\mathcal{R}_t^{(k)} = \hat{x}_{t-1}^{(k)} - \mathcal{F}_t^{(i)}(\hat{x}_t^{(k)}, \cdots, \hat{x}_{t+i}^{(k)})$ indicates a residual term to be optimized; and $\mathcal{G}^{(k)} = \left(\mathcal{J}^{(k)}\right)^{-1}$ indicates the inverse of Jacobian matrix $\mathcal{J}^{(k)} = \frac{\partial \mathcal{R}_{0:T}^{(k)}}{\partial \hat{x}_{0:T}}$.

**Choices of Approximating Jacobian Matrix $\mathcal{J}^{(k)}$.** A key strategy for accelerating parallel sampling solvers is to efficiently approximate the Jacobian matrix in the Newton update step, rather than computing the full matrix. Previous methods have employed distinct approximation schemes: *ParaDIGMS* (Shih et al., 2024) uses Picard iteration, a fixed-point method that avoids explicit Jacobian computation. This approach is equivalent to approximating the Jacobian of the system as the identity matrix as $\mathcal{J}^{(k)} \approx I$, simplifying the expensive Newton step into a computationally cheap update. *ParaTAA* (Tang et al., 2024a) adapts Anderson Acceleration to the problem's causal structure. Standard acceleration can produce a dense update matrix, which allows well-converged variables to be corrupted by those that have not yet converged. ParaTAA resolves this by enforcing a block upper triangular structure on its update matrix, preserving stability by respecting the natural flow of information in the diffusion process. *ParaSolver* (Lu et al., 2025)

formulates the problem to have an Jacobian matrix consists of identity blocks on the main diagonal and non-zero blocks only on the sub-diagonal, which reduces the computational and memory costs of each solver iteration. However, current works all face generalization challenges when scaling to larger scale generation. This leads to the following question that we aim to explore in this work:

*Can we dynamically control the sparsity of the Jacobian in parallel diffusion samplers to achieve a better stability–efficiency trade-off, thereby enabling scalable high-dimensional and multi-modal generation?*

**Our Contributions.** Following the research question, we introduce **ROPA** (**RO**bust **PA**rallel diffusion), a novel framework that achieves a superior balance between the efficiency of parallel solving and numerical stability, which scales the application of parallel sampling to complex tasks like video generation. Our key contributions are:

**(a) Scaling To High-Dimensional Generation**. Our analysis in Section 2 identifies a score-field stiffness mechanism that limits the scalability of existing parallel diffusion samplers. In particular, regions with large local score curvature induce large drift Jacobians, and sparse or fixed-band Jacobian approximations further introduce truncation error in the parallel residual system. Together, these effects worsen the conditioning of Newton-type updates and explain the unstable convergence observed in high-dimensional image and video generation.

Current parallel sampling methods, as shown in Figure 1, can therefore struggle when the local score curvature and the banded-Jacobian truncation error jointly make the residual Jacobian ill-conditioned. This leads to unreliable convergence within practical iteration budgets and may cause trajectories to drift from the sequential reference path. ROPA addresses this issue by adaptively regulating the effective Jacobian bandwidth and damping unstable updates, improving numerical stability while preserving the computational efficiency of sparse parallel solving.

**b)** We propose *Residual-Driven Adaptive Jacobian Bandwidth Control*. Specifically, we translate local residual diagnostics into on-the-fly control of the solver's coupling structure. At each iteration, the method widens the lookahead where residuals suggest local truncation or linearization error, and prunes it elsewhere to preserve sparse parallel computation. When diagnostics flag unstable updates, an adaptive damping mechanism moderates the step size—behaving like fast Newton updates in well-conditioned regions and shifting toward conservative updates near ill-conditioning. Together, adaptive bandwidth, damped least-squares updates, and score-aligned preconditioning improve the conditioning–efficiency trade-off of the banded resid-

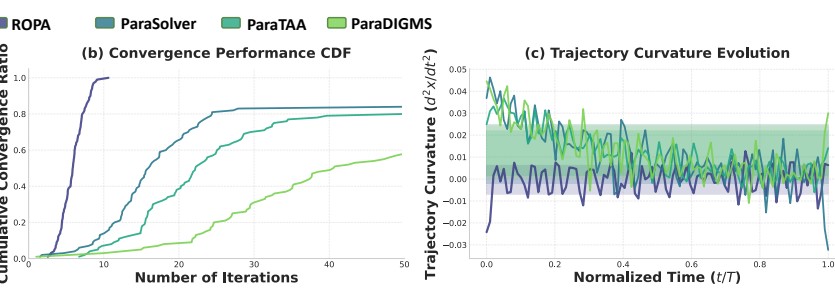

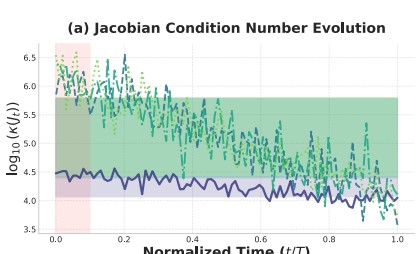

*Figure 1.* ROPA Performance Analysis. (a) Estimated Jacobian conditioning showing improved numerical stability. (b) Convergence CDF demonstrating faster and more reliable residual reduction. (c) Trajectory-curvature diagnostic showing reduced deviation from the reference sampling path. Shaded regions represent mean $\pm 1\sigma$ uncertainty bounds. The red shaded area in (a) indicates a locally stiff region where numerical challenges are most severe.

ual solve, enabling training-free parallel sampling for large image and video diffusion models.

**c)** Extensive experiments demonstrate substantial speedups on Stable Diffusion-v3.5, FLUX, HunyuanVideo, Wan2.1 and CogVideoX while maintaining FID and CLIP scores.

## 2. A Unified Geometric Analysis of Parallel Sampling Instability in Diffusion Models

We establish a framework linking score-field curvature, discretization-induced truncation error, and numerical stability in parallel diffusion sampling. This framework explains why fixed sparse Jacobian approximations can become unstable in locally stiff regions and motivates adaptive Jacobian-bandwidth control.

### 2.1. Geometric Foundations of Score Stiffness and Density Curvature

We use the term *curvature* in a probabilistic rather than purely Riemannian sense. Let

$$s_t(x) := \nabla_x \log p_t(x)$$

denote the score field of the smoothed density at diffusion time $t$. We define the local score-curvature magnitude by

$$C_t(x) := \|\nabla_x s_t(x)\|_2 = \|\nabla_x^2 \log p_t(x)\|_2.$$

Unlike intrinsic curvature of the data manifold, $C_t(x)$ measures how rapidly the score changes in the ambient space. This quantity directly controls the stiffness of the reverse drift and, through the discretized residual system, the conditioning of the parallel solver.

The core challenge stems from the data manifold's intrinsic curvature properties. Let $\mathcal{M} \subset \mathbb{R}^d$ denote the support of $p_0(x)$, with curvature characterized by the score Hessian $\mathcal{H}(x) = \nabla_x^2 \log p(x)$. The eigenvalues of $\mathcal{H}(x)$ quantify how sharply the density bends in different directions. A large ratio between the largest and smallest eigenvalues, corresponding to high anisotropy, means that the score changes

very quickly along some directions but slowly along others, this is precisely the notion of stiffness that leads to ill-conditioned Jacobians in our parallel residual system.

**Assumption 2.1** (Local Score-Curvature Regularity). For each diffusion time $t \in (0, T]$, the smoothed density $p_t$ admits a continuously differentiable score field $s_t(x) = \nabla_x \log p_t(x)$. Define

$$C_t(x) := \|\nabla_x s_t(x)\|_2.$$

We assume $C_t(x)$ is locally bounded and locally Lipschitz along sampled trajectories. Moreover, large values of $C_t(x)$ may occur only on a measurable subset of the trajectory domain corresponding to locally stiff score dynamics.

**Theorem 2.2** (Score Curvature and Denoiser Jacobian). *Let $r_\star(x, t) = \mathbb{E}[x_0 \mid x_t = x]$ denote the exact posterior denoiser. Suppose the denoiser satisfies the $C^1$-approximation condition*

$$\|J_{r_\theta}(x, t) - J_{r_\star}(x, t)\|_2 \le \varepsilon_J.$$

*Under the standard Gaussian perturbation model, $r_\star$ admits a Tweedie-type form $r_\star(x, t) = a_t x + b_t s_t(x)$ for scalar coefficients $a_t, b_t$. Therefore,*

$$\|J_{r_\theta}(x, t)\|_2 \le |a_t| + |b_t|\, C_t(x) + \varepsilon_J.$$

*Moreover, whenever the leading score-curvature direction is not canceled by the identity component,*

$$\|J_{r_\theta}(x, t)\|_2 \ge |b_t|\, C_t(x) - |a_t| - \varepsilon_J.$$

*Thus, large score curvature can induce a large denoiser Jacobian and hence stiff reverse dynamics.*

### 2.2. Discretization-Induced Instability

The residual system in parallel sampling is defined as $\mathcal{R}_n^{(k)} = \hat{x}_{t_{n-1}}^{(k)} - \mathcal{F}_{t_n}^{(i)}(\hat{x}_{t_n}^{(k)}, \dots, \hat{x}_{t_{n+i}}^{(k)})$ per Eq. 6, where for Euler integration:

$$\mathcal{F}_{t_n}^{(i)}(x_{t_n}, \dots, x_{t_{n+i}}) = x_{t_n} - \Delta\varphi(x_{t_n}, t_n), \quad \Delta = t_{n+1} - t_n. \tag{7}$$

This discretization introduces gaps between continuous and discrete dynamics:

**Theorem 2.3** (Conditioning of a Banded Parallel Residual Jacobian). *Let $\mathcal{J}_b^{(k)}$ be the block-banded Jacobian used by the parallel residual solver with bandwidth b. Write*

$$\mathcal{J}_b^{(k)} = I + hA^{(k)} + E_b^{(k)},$$

*where $A^{(k)}$ collects the local drift-Jacobian blocks and $E_b^{(k)}$ denotes the truncation error caused by the finite bandwidth. Assume*

$$\|A^{(k)}\|_2 \le L_t, \qquad \|E_b^{(k)}\|_2 \le \tau_b, \qquad hL_t + \tau_b < 1.$$

*Then*

$$\kappa(\mathcal{J}_b^{(k)}) \le \frac{1 + hL_t + \tau_b}{1 - hL_t - \tau_b}.$$

*For the reverse drift $\varphi(x, t) = f(t)x - g^2(t)s_t(x)$, we have*

$$L_t \lesssim |f(t)| + g^2(t)C_t(x).$$

*Consequently, large score curvature $C_t(x)$ and large truncation error $\tau_b$ jointly worsen the conditioning of the banded residual Jacobian.*

This yields the *score-curvature numerical cascade*: large $C_t(x)$ increases the local drift stiffness $L_t$, while an insufficient Jacobian bandwidth increases the truncation error $\tau_b$. When $hL_t + \tau_b$ approaches one, the condition-number bound deteriorates and Newton-type updates become unreliable. Thus, instability is governed by local score-field stiffness and discretization error, rather than by diffusion time alone.

## 2.3. Trajectory Geometry and Mode-Interpolation Risk

The stability loss also manifests as trajectory deviation from the reference sequential sampler. Following Chen & Muñoz Ewald (2023), we use trajectory quasi-linearity, $\|d^2x/dt^2\|_2 \le \varepsilon$, as a diagnostic of local numerical difficulty. We obtain the following backward-forward error interpretation:

**Corollary 2.4** (Numerical Stability & Reference-Trajectory Deviation). *Under the same hypotheses as Theorem 2.3, let $\hat{x}^{(k+1)}$ be the iterate returned by one inexact Newton step and let $x_{\text{ref}}^{(k+1)}$ denote the corresponding update obtained with a well-conditioned reference Jacobian. Then, up to higher-order residual terms,*

$$\|\hat{x}^{(k+1)} - x_{\text{ref}}^{(k+1)}\|_2 \lesssim \kappa(\mathcal{J}_b^{(k)})\|\mathcal{J}_b^{(k)-1}\mathcal{R}^{(k)}\|_2 + \mathcal{O}(\|\mathcal{R}^{(k)}\|_2^2) \tag{8}$$

*Hence, poor conditioning can amplify residual errors and cause the parallel trajectory to deviate from the sequential reference path.*

This gives a conservative explanation for mode-interpolation risk: regions with rapidly varying score fields are important for separating modes, but they also make the residual system harder to solve with a fixed sparse Jacobian. Such regions often appear near low-density transition areas between modes:

**Corollary 2.5** (Boundary Sensitivity in Low-Temperature Mixtures). *Consider a low-temperature mixture*

$$p_\tau(x) = \sum_m w_m \exp(-E_m(x)/\tau),$$

*where $\tau > 0$ controls the sharpness of the modes. In transition regions where multiple components have comparable weights, the score curvature $C_\tau(x) = \|\nabla_x^2 \log p_\tau(x)\|_2$ can grow polynomially in $1/\tau$. Consequently, sharp low-density transition regions can increase local drift stiffness and worsen the conditioning bound in Theorem 2.3.*

Theoretical results suggest that the condition number $\kappa(\mathcal{J}_b)$—modulated by score-field stiffness, finite-band truncation error, and discretization step size—is a useful diagnostic connecting numerical stability with trajectory fidelity. To translate this insight into a practical sampler, we introduce three solver-control principles that improve the conditioning-efficiency trade-off of Newton-type parallel updates without requiring explicit curvature estimation. Each principle is summarized as below.

**Takeaways 2.6** (Damped Updates for Safety). At iteration $k$, we stabilize the Newton step with a damped least-squares update

$$\left((\mathcal{J}^{(k)})^\top \mathcal{J}^{(k)} + \lambda_k I\right)\Delta x^{(k)} = -(\mathcal{J}^{(k)})^\top \mathcal{R}^{(k)},$$

where $\lambda_k > 0$ is increased when the candidate update fails to sufficiently reduce the residual. This Levenberg–Marquardt-style damping limits the update gain in ill-conditioned directions and provides a practical descent safeguard for non-symmetric residual Jacobians.

**Takeaways 2.7** (Adaptive Sparsity for Efficiency). Let $\mathcal{S}_k$ be a block-band sparsity pattern with bandwidth $b_k$, and let $\tau_{b_k}$ denote the corresponding off-band truncation error. Theorem 2.3 shows that the conditioning is controlled by $hL_t + \tau_{b_k}$. Therefore, increasing $b_k$ is useful only where it substantially reduces $\tau_{b_k}$, while small residual regions can safely use narrower bands. This motivates residual-driven bandwidth adaptation with per-step cost proportional to the selected bandwidth.

**Takeaways 2.8** (Score-Aligned Low-Rank Preconditioning). When the residual is strongly aligned with the score direction, ROPA applies a rank-one score-aligned preconditioner that reduces the update gain along this potentially stiff direction. This does not require Hessian eigendecomposition; it only uses the available score vector and is activated conditionally by a residual-score alignment test.

**Summary** The local score-curvature magnitude controls drift stiffness (Theorem 2.2), while discretization and finite-band approximations affect the conditioning of the residual Jacobian (Theorem 2.3). Poor conditioning can amplify residual errors and cause the parallel trajectory to deviate from the sequential reference path (Corollary 2.4), especially in sharp transition regions between modes (Corollary 2.5). ROPA's adaptive mechanisms are designed to reduce this numerical risk by controlling bandwidth, damping, and score-aligned update gain. See Appendix B for proofs.

# 3. Robust Parallel Diffusion Sampling via Adaptive Jacobian Sparsity

Building on the score-curvature numerical cascade in Section 2, we aim to regulate the conditioning of the banded residual Jacobian $\mathcal{J}_b$. While Theorem 2.2 links score curvature $C_t(x)$ to local drift stiffness, explicitly estimating $C_t(x)$ at inference time is expensive. Theorem 2.3 instead suggests a practical control target: the combined effect of drift stiffness and bandwidth truncation, summarized by $hL_t + \tau_b$.

ROPA therefore uses the residual norm as a cheap diagnostic of local linearization or truncation failure. This diagnostic drives three lightweight solver controls: (i) adaptive bandwidth control, which allocates wider residual couplings only where needed; (ii) damped least-squares updates, which prevent unstable Newton steps; and (iii) score-aligned low-rank preconditioning, which moderates updates when residuals align with the score direction.

Let $N := T + 1$ denote the total number of discrete time points on the grid $\{t_0, \ldots, t_T\}$.

## 3.1. Dynamic Residuals with Adaptive Jacobian Bandwidth

At each grid index $i \in \{1, \ldots, T\}$ the algorithm selects a forward-looking window width $w_i \in \{1, \ldots, w_{\max}\}$ and forms the residual

$$\mathcal{R}_i^{(w_i)}(\hat{\mathbf{x}}) = \hat{\mathbf{x}}_{t_{i-1}} - \Psi_i^{(w_i)}\big(\hat{\mathbf{x}}_{t_i}, \ldots, \hat{\mathbf{x}}_{t_{i+w_i-1}}\big), \quad (9)$$

where $\Psi_i^{(w_i)}$ approximates the integral operator $\Phi(t_{i-1}, t_i, \hat{\mathbf{x}}_{t_i})$ by means of an explicit $w_i$-step integrator (e.g., Euler, DDIM, or a higher-order variant). This look-ahead construction yields an upper-banded Jacobian:

$$\frac{\partial \mathcal{R}_i^{(w_i)}}{\partial \hat{\mathbf{x}}_{t_j}} = \begin{cases} \mathbf{I}_d, & j = i-1, \\ -\dfrac{\partial \Psi_i^{(w_i)}}{\partial \hat{\mathbf{x}}_{t_j}}, & i \leq j \leq i + w_i - 1, \\ \mathbf{0}, & \text{otherwise.} \end{cases} \quad (10)$$

We view this finite-window Jacobian as a banded approximation $\mathcal{J}_b = \mathcal{J}_{\text{full}} + E_b$, where $E_b$ collects the off-band coupling terms omitted by the current window width. For first-order integrators this structure often yields favorable block-row dominance. Higher-order schemes may weaken that dominance; a locally scaled damping parameter $\lambda_{\text{damp},i}$, described in Section 3.2, is used as a practical safeguard in that case.

**Adaptive bandwidth control.** During Newton iteration $k$, the algorithm evaluates local residual norms

$$e_i^{(k)} = \big\|\mathcal{R}_i^{(w_i^{(k)})}(\hat{\mathbf{x}}^{(k)})\big\|_2 \quad (11)$$

and their global mean

$$\bar{e}^{(k)} = N^{-1} \sum_{i=0}^{T} e_i^{(k)}. \quad (12)$$

Following **Theorem 2.3**, insufficient bandwidth enters the residual Jacobian through the truncation term $\tau_b$. Since $\tau_b$ is not directly available during sampling, we use the local residual norm $e_i^{(k)}$ as a cheap diagnostic of local linearization or truncation failure. To reduce this error where it is most likely to affect conditioning, we dynamically adjust the window widths:

$$w_i^{(k+1)} = \begin{cases} \min\{w_i^{(k)} + 1, w_{\max}\}, & e_i^{(k)} > \alpha\bar{e}^{(k)}, \\ \max\{w_i^{(k)} - 1, 1\}, & e_i^{(k)} < \beta\bar{e}^{(k)}, \\ w_i^{(k)}, & \text{otherwise,} \end{cases} \quad (13)$$

with default parameters $\alpha = 1.5$ and $\beta = 0.7$.

By *densifying* the block-banded Jacobian only in high-error regions, the update rule reduces the local truncation error $\tau_b$ whenever the off-band couplings are the dominant source of residual error. According to Theorem 2.3, this improves the conditioning bound by decreasing the term $hL_t + \tau_b$. Conversely, pruning low-error regions keeps the effective bandwidth small, preserving the efficiency of sparse parallel solving.

## 3.2. Score-Aligned Low-Rank Preconditioning

Adaptive bandwidth reduces truncation error caused by omitted temporal couplings, while damping controls the magnitude of Newton-type updates. However, in some iterations the residual is concentrated along the score direction, indicating that the current update is dominated by a locally stiff denoising direction. ROPA detects this case by monitoring the residual-score alignment

$$a_i^{(k)} = \frac{|\langle \mathcal{R}_i^{(k)}, \mathbf{s}_\theta^{(k)} \rangle|}{\|\mathcal{R}_i^{(k)}\|_2 \|\mathbf{s}_\theta^{(k)}\|_2 + \delta_s}, \quad (14)$$

where $\delta_s > 0$ avoids numerical instability. When $a_i^{(k)} > \gamma$, we apply a score-aligned rank-one preconditioner. Let

$$u_i^{(k)} = \frac{\mathbf{s}_\theta^{(k)}}{\|\mathbf{s}_\theta^{(k)}\|_2 + \delta_s}.$$

We define

$$\mathbf{P}_i^{(k)} = \mu_{\perp,i}^{(k)} I + \left(\mu_{\|,i}^{(k)} - \mu_{\perp,i}^{(k)}\right) u_i^{(k)} (u_i^{(k)})^\top, \quad (15)$$

where

$$\mu_{\|,i}^{(k)} = \frac{1}{\lambda_{\text{damp},i}^{(k)} + g^2(t_i)\|\mathbf{s}_\theta^{(k)}\|_2^2}, \qquad \mu_{\perp,i}^{(k)} = \frac{1}{\lambda_{\text{damp},i}^{(k)}}.$$

Thus the update is more conservative along the score-aligned direction when the score norm is large, while the orthogonal subspace is controlled by the same damping parameter.

For affected grid indices, the local Jacobian block is preconditioned as

$$\mathbf{B}_i^{(k)} = \begin{bmatrix} -\mathbf{P}_i^{(k)} & \mathbf{I}_d \end{bmatrix}. \quad (16)$$

This construction uses only the score vector already computed by the diffusion model, so the additional memory remains $\mathcal{O}(Nw_{\text{max}}d)$. It should be viewed as a conditional score-aligned preconditioner rather than an explicit inverse-Hessian estimate; its role is to reduce update amplification when residuals are aligned with a stiff score direction.

# 4. Experiments

## 4.1. Setups

**Models.** For video generation, we benchmark on three state-of-the-art large video diffusion models: Hunyuan-Video (Kong et al., 2024) and CogVideoX1.5-5B (Yang et al., 2024b). For each model, we generate videos with prompts in VBench (Huang et al., 2024) strictly following VBench evaluation protocol. We consider two image diffusion models for image generation, Stable Diffusion 3.5 Large (Esser et al., 2024) and FLUX (Labs, 2024), as the backbone. Following previous works (Shih et al., 2024; Selvam et al., 2024), we sample 1000 prompts from COCO2017 captions dataset as the test bed. We use $N = 50$ diffusion steps by default, with more investigations on $N$ in Appendix.

**Algorithms.** We benchmark our proposed algorithm, ROPA, against five key baselines: (1) *Sequential Sampling*, the standard non-parallel approach which serves as the reference for performance speedups; (2) *ParaDiGMS* (Shih et al., 2023), a foundational parallel method utilizing Picard (fixed-point) iteration; (3) *ParaTAA* (Tang et al., 2024b), which accelerates convergence by applying Triangular Anderson Acceleration (TAA) to a dense nonlinear system; (4) *ParaSolver* (Lu et al., 2025), a highly efficient method that combines a

quasi-Newton solver with a sparse, banded system structure; and (5) *CHORDS* (Han et al., 2025), a parallel framework designed for robust and stable convergence.

**Hyperparameter Settings.** **Damping $\lambda$:** We use a Levenberg–Marquardt-style rule: starting from $10^{-3}$, $\lambda$ is increased by a factor of 2 until the candidate update is sufficiently small, $\|\delta\hat{x}\|/\|\mathcal{R}\| < 0.3$, or yields residual decrease. This limits update amplification in ill-conditioned directions. **Prune factor $\eta$:** Set to 0.1 for images, 0.2 for videos (robust in $[0.05, 0.3]$), with threshold $\gamma$ explicitly using $\|S_\theta\|^2$ from the introduction. **Adaptation thresholds:** $\alpha = 1.5$, $\beta = 0.7$ provide optimal sparsity-stability balance, directly addressing the convergence degradation near $t \to 0$ observed in the introduction. This configuration enables stable high-fidelity generation at scale while maintaining $O(N)$ parallelism— even in high-curvature regions where existing methods fail, as empirically demonstrated in the introduction's Figures 1.

**Settings.** We run experiments using 8 * H200 GPUs, each with 140GB of memory. We use model-specific classifier-free guidance scales, as listed in Appendix A. The window-scaled variant halves the number of synchronization rounds compared with a fixed $\lambda$. For all algorithms, we use the same stopping threshold $\varepsilon_t = \tau^2 g^2(t)d$ with $\tau = 10^{-3}$, and initialize all variables with standard Gaussian Distribution and warming-up steps set as 3.

**Evaluation.** For both video and image models, we report *Time per sample* that refers to the average wall-clock time used to generate one sample. *Speedup* that refers to the relative speedup compared with sequential solve, measured by the number of sequential network forward calls. Notice that this will be slightly different from the measurement or the wall-clock. In terms of generation quality, we report average of diverse *Quality Scores* (Clarity, Aesthetic, Motion, Dynamic, Semantic, Anatomy, Identity) normalized using the same numerical system as the standard quality metric following the VBench evaluation protocol (Huang et al., 2024) for video generation, and *CLIP Score* (Hessel et al., 2021) evaluated using ViT-g-14 (Radford et al., 2021) for the image generation. We also report *Latent RMSE* under both cases that measures the Rooted MSE between the returned latent of the algorithm and that of the sequential solver. Notice that a lower latent RMSE indicates lower sampling error, with sequential solve being the oracle.

## 4.2. Main Results

**Video diffusion acceleration** Our proposed ROPA, demonstrates a clear superiority across all tested models in Table. 1. At the highest level of parallelism with 8 cores, ROPA achieves remarkable speedups ranging from $2.5\times$ to $2.9\times$. On the HunyuanVideo model, it reduces the generation time from 378.6s to just 131.8s, a $2.9\times$ acceleration. This significant performance gain is achieved without any meaningful

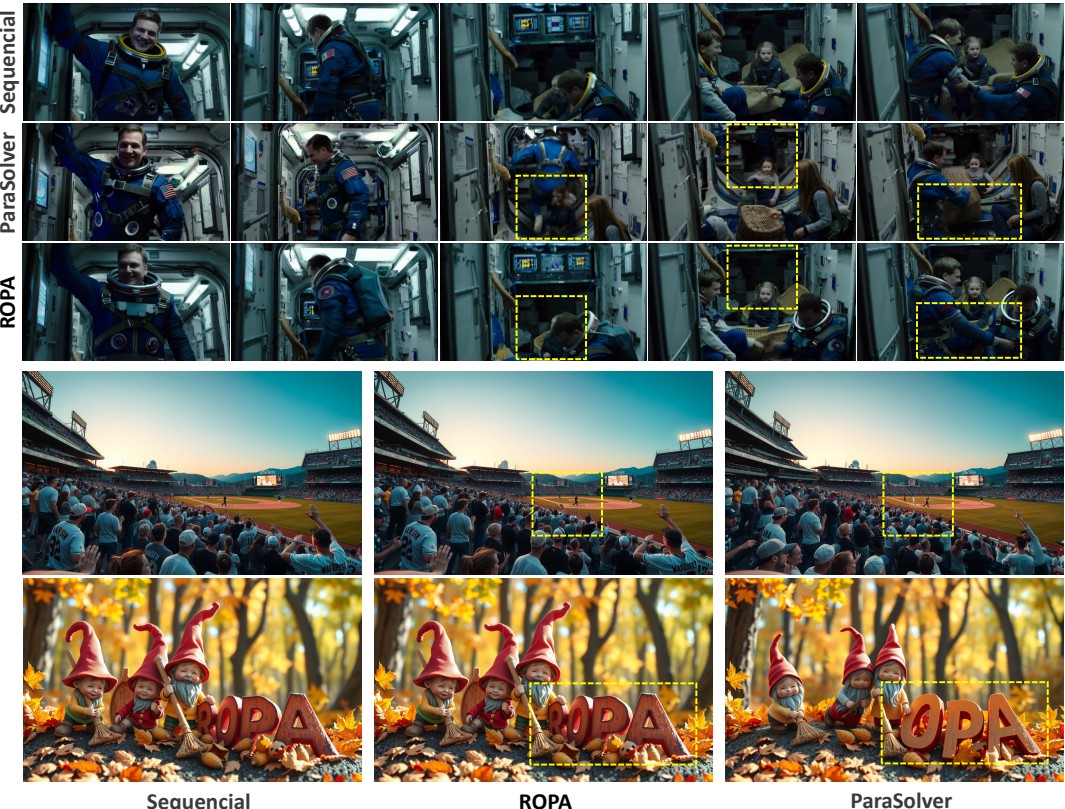

*Figure 2.* Quality Comparison of ROPA and baselines on HunyuanVideo and Flux models.

degradation in output quality. The VBench Quality score remains exceptionally stable, 73.6% for HunyuanVideo vs. 73.8% for the sequential baseline, and the Latent RMSE is kept to a minimum. Notably, ROPA's Latent RMSE of 0.055 is not only competitive with the best-performing baselines but is also nearly three times lower than the 0.189 error of *ParaDIGMS*, highlighting its ability to accelerate sampling while preserving high fidelity.

**Image diffusion acceleration.** The benchmark results of image generation are presented in Table. 2. Similar to video generation, ROPA maintains significant speedups across different numbers of cores on image diffusion models, achieving up to 39% speedup improvement over the strongest baseline at four cores and reaching up to 2.3× speedup with eight cores. Notice that this is obtained with the lowest latent RMSE and negligible change in CLIP Score, suggesting the superiority of ROPA.

**Higher robustness brings lower number of function evaluations.** ROPA's core advantage lies in its numerical robustness, which directly translates to a lower required Number of Function Evaluations (NFE) for convergence. The adaptive damping and geometry-aware preconditioning mechanisms allow ROPA to handle the stiff, high-curvature regions of the sampling trajectory where simpler methods

like *ParaDIGMS* struggle. As demonstrated in our experiments, while baselines often require additional iterations or fail to converge, ROPA consistently converges in an average of 8-12 outer Newton iterations. This stability ensures a predictable and efficient path to a high-fidelity solution, effectively minimizing the total computational work needed.

**Reference-trajectory fidelity under convergence.** Figure 2 shows that ROPA reaches a lower-error trajectory within the same practical NFE budget. Even when baselines are allowed additional refinement steps, their latent RMSE often plateaus above ROPA, indicating that numerical instability can lead to persistent deviation from the sequential reference path. This observation is consistent with Corollary 2.4: poor conditioning amplifies residual errors, whereas ROPA reduces this amplification through adaptive bandwidth, damping, and score-aligned preconditioning.

### 4.3. Ablation Study

**Effect of Main Components.** To validate the contributions of each component in ROPA, we conducted an ablation study, systematically deactivating key mechanisms. The results, summarized in Table. 3, confirm that all parts are integral to performance. Full ROPA serves as our baseline. The Ada-J variant isolates adaptive bandwidth control, while

*Table 1.* Benchmark results of parallel diffusion methods on video diffusion models using VBench. We evaluate on three video diffusion models with the number of cores $K$ set to 2, 4 and 8. Our approach achieves the highest speedup without measurable quality degradation.

| | | Num Core = 2 | | | | Num Core = 4 | | | | Num Core = 8 | | | |
| --- | --- | --- | --- | --- | --- | --- | --- | --- | --- | --- | --- | --- | --- |
| | | Time(s) | Speedup | Quality$_V$ | RMSE$_L$ | Time(s) | Speedup | Quality$_V$ | RMSE$_L$ | Time(s) | Speedup | Quality$_V$ | RMSE$_L$ |
| HunyuanVideo | Sequential | 378.6 | - | **73.8%** | - | 378.6 | - | **73.8%** | - | 378.6 | - | **73.8%** | - |
| | CHORDS | 292.3 | 1.3 | 73.6% | 0.188 | 185.5 | 2.0 | 73.7% | 0.182 | 156.0 | 2.4 | 73.7% | 0.185 |
| | ParaDIGMS | 313.3 | 1.2 | 73.7% | 0.190 | 293.1 | 1.3 | 73.6% | 0.175 | 271.8 | 1.4 | 73.6% | 0.189 |
| | ParaTAA | 318.6 | 1.2 | 73.6% | 0.055 | 207.0 | 1.8 | 73.6% | 0.055 | 157.1 | 2.4 | 73.6% | 0.055 |
| | ParaSolver | 287.5 | 1.3 | 73.5% | **0.051** | 208.1 | 1.8 | 73.5% | **0.049** | 164.7 | 2.3 | 73.5% | **0.052** |
| | **ROPA (Ours)** | **232.8** | **1.6** | 73.6% | 0.054 | **177.9** | **2.1** | 73.6% | 0.053 | **131.8** | **2.9** | 73.6% | 0.055 |
| Wan2.1 | Sequential | 471.2 | - | **74.7%** | - | 471.2 | - | **74.7%** | - | 471.2 | - | **74.7%** | - |
| | CHORDS | 362.8 | 1.3 | 74.5% | 0.082 | 274.9 | 1.7 | 74.6% | 0.076 | 197.0 | 2.4 | 74.6% | 0.079 |
| | ParaDIGMS | 395.1 | 1.2 | 74.5% | 0.077 | 332.6 | 1.4 | 74.6% | 0.070 | 279.6 | 1.7 | 74.6% | 0.084 |
| | ParaTAA | 338.2 | 1.4 | 74.5% | 0.030 | 312.9 | 1.5 | 74.5% | 0.028 | 202.1 | 2.3 | 74.5% | 0.028 |
| | ParaSolver | 340.2 | 1.4 | 74.5% | **0.025** | 293.2 | 1.6 | 74.5% | 0.024 | 185.2 | 2.5 | 74.5% | **0.026** |
| | **ROPA (Ours)** | **274.0** | **1.7** | 74.5% | 0.027 | **250.8** | **1.9** | 74.5% | **0.021** | **169.1** | **2.8** | 74.5% | 0.030 |
| CogVideoX1.5 | Sequential | 464.5 | - | **71.3%** | - | 464.5 | - | **71.3%** | - | 464.5 | - | **71.3%** | - |
| | CHORDS | 389.5 | 1.2 | 71.0% | 0.132 | 246.3 | 1.9 | 71.1% | 0.125 | 221.5 | 2.1 | 71.0% | 0.129 |
| | ParaDIGMS | 390.9 | 1.2 | 71.0% | 0.146 | 356.3 | 1.3 | 71.0% | 0.119 | 290.7 | 1.6 | 70.9% | 0.174 |
| | ParaTAA | 359.9 | 1.3 | 70.9% | 0.043 | 388.0 | 1.2 | 70.9% | 0.043 | 224.1 | 2.1 | 70.9% | 0.043 |
| | ParaSolver | 332.4 | 1.4 | 71.0% | **0.040** | 386.9 | 1.2 | 71.1% | **0.039** | 207.5 | 2.2 | 71.0% | **0.041** |
| | **ROPA (Ours)** | **307.5** | **1.5** | 71.1% | 0.041 | **219.5** | **2.1** | 71.2% | 0.041 | **182.4** | **2.5** | 71.2% | 0.042 |

*Table 2.* Benchmark results of parallel diffusion methods on latent image diffusion models. We evaluate two models with 1000 prompts from the COCO2017 captions dataset. Our approach achieves the highest speedup without measurable quality degradation.

| | | Num Core = 2 | | | | Num Core = 4 | | | | Num Core = 8 | | | |
| --- | --- | --- | --- | --- | --- | --- | --- | --- | --- | --- | --- | --- | --- |
| | | Time(s) | Speedup | CLIP | RMSE$_L$ | Time(s) | Speedup | CLIP | RMSE$_L$ | Time(s) | Speedup | CLIP | RMSE$_L$ |
| SD-3.5-Large | Sequential | 10.3 | - | 37.4 | - | 10.3 | - | 37.4 | - | 10.3 | - | 37.4 | - |
| | ParaDIGMS | 7.6 | 1.4 | 37.2 | 0.440 | 7.7 | 1.3 | 37.4 | 0.346 | 7.1 | 1.5 | **37.4** | 0.342 |
| | ParaSolver | 6.8 | 1.5 | **37.4** | 0.234 | 9.4 | 1.1 | 37.4 | 0.294 | 5.8 | 1.8 | 37.3 | 0.324 |
| | **ROPA (Ours)** | **6.3** | **1.6** | **37.4** | **0.141** | **5.8** | **1.8** | **37.4** | **0.220** | **5.2** | **2.0** | **37.4** | **0.224** |
| Flux | Sequential | 11.2 | - | 37.4 | - | 11.2 | - | 37.4 | - | 11.2 | - | 37.4 | - |
| | ParaDIGMS | 8.1 | 1.4 | 37.4 | 0.249 | 7.2 | 1.6 | 37.4 | **0.121** | 7.3 | 1.5 | 37.4 | 0.313 |
| | ParaSolver | 6.4 | 1.7 | 37.3 | 0.270 | 6.6 | 1.7 | 37.4 | 0.166 | 5.5 | 2.0 | 37.4 | 0.150 |
| | **ROPA (Ours)** | **5.8** | **1.9** | 37.4 | **0.154** | **5.3** | **2.1** | 37.4 | 0.143 | **4.8** | **2.3** | **37.4** | **0.120** |

*Table 3.* Evaluation of main components and compatibility of other acceleration methods at $K = 4$. Ada-J denotes adaptive Jacobian bandwidth control, Curv-C denotes score-aligned low-rank preconditioning, and SA denotes SpargeAttention.

| | FLUX | | | | HunyuanVideo | | | |
| --- | --- | --- | --- | --- | --- | --- | --- | --- |
| | Time(s) | Speedup | CLIP | RMSE$_L$ | Time(s) | Speedup | Quality$_V$ | RMSE$_L$ |
| Sequential | 11.2 | - | 37.4 | - | 378.6 | - | 73.8 | - |
| w/ Ada-J | 8.9 | 1.3 | 37.4 | 0.145 | 252.4 | 1.5 | 73.7 | 0.062 |
| w/ Curv-C | 9.2 | 1.2 | 37.4 | 0.142 | 270.3 | 1.4 | 73.8 | 0.058 |
| w/ SA | 6.8 | 1.6 | 36.8 | 0.180 | 210.5 | 1.8 | 72.1 | 0.095 |
| w/ ToCa | 7.1 | 1.6 | 36.9 | 0.175 | 220.3 | 1.7 | 72.3 | 0.088 |
| ROPA (Ours) | 5.3 | **2.1** | 37.4 | 0.143 | 177.9 | **2.1** | 73.6 | 0.053 |
| w/ SA | 4.8 | 2.3 | 36.9 | 0.165 | 158.2 | 2.4 | 72.8 | 0.078 |
| w/ ToCa | 5.0 | 2.2 | 37.0 | 0.160 | 162.5 | 2.3 | 73.0 | 0.072 |

Curv-C isolates the score-aligned low-rank preconditioning component. Removing adaptive bandwidth increases the number of inner linear-solver iterations, since the banded Jacobian fails to capture important local couplings. Removing score-aligned preconditioning mainly affects difficult regions where the residual is concentrated along the score direction. These results show that bandwidth adaptation and score-aligned preconditioning contribute complementary

gains, while damping is kept as a shared safeguard across solver variants.

**Compatibility with other Diffusion Acceleration Scheme.** ROPA's algorithmic improvements are complementary to structural-level optimizations—such as training-free sparse attention SpargeAttention (Zhang et al., 2025)) and Attention Token-wise Caching ToCa (Zou et al., 2025)—as illustrated in Table. 3. To this end, we integrated ROPA and the baseline methods with a standard attention cache and re-evaluated their performance. Our results show that while attention caching reduced the wall-clock time per function evaluation across all methods, ROPA retained its relative speedup advantage. For instance, on HunyuanVideo with caching enabled, ROPA remained 2.4× faster than the sequential baseline. This confirms that ROPA delivers orthogonal, algorithmic-level acceleration by reducing the NFE, which multiplies synergistically with techniques.

**Empirical verification of residual diagnostics.** To validate the residual-driven bandwidth rule, we estimate the local stiffness $L(x_t)$ along sampled trajectories and compare

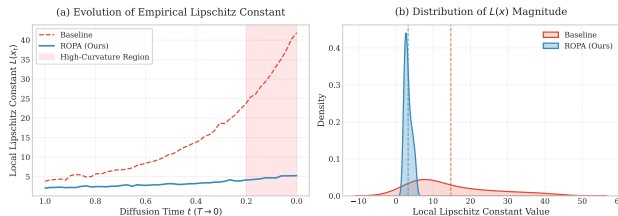

*Figure 3.* **Empirical Analysis of Local Stiffness and Residual Diagnostics.** (Left) Evolution of the estimated local stiffness $L(x_t)$ along the sampling trajectory. Baseline methods show larger stiffness and residual spikes in difficult regions, while ROPA reduces the effective residual through adaptive bandwidth and damping. (Right) Distribution of estimated $L(x_t)$ values. The bounded empirical range supports the local regularity assumptions used in our conditioning analysis.

*Table 4.* Adaptive bandwidth statistics across diffusion time. We report mean/median $w_i$ and the fraction of steps with $w_i \geq 5$.

| Model / region | Early | Middle | Late stiff |
|---|---|---|---|
| FLUX mean / median | 1.3 / 1 | 2.5 / 2 | 6.6 / 7 |
| FLUX $\Pr(w_i \geq 5)$ | 0% | 10% | 85% |
| Hunyuan mean / median | 1.5 / 1 | 2.9 / 3 | 7.1 / 7 |
| Hunyuan $\Pr(w_i \geq 5)$ | 0% | 17% | 91% |

it with the residual profile used by ROPA. We approximate the spectral norm of the denoiser Jacobian by finite differences:

$$L(x_t) \approx \max_{v \sim \mathcal{N}(0,I)} \frac{\|\epsilon_\theta(x_t + \delta v, t) - \epsilon_\theta(x_t, t)\|_2}{\|\delta v\|_2},$$

where $\delta = 10^{-4}$. As shown in Figure 3, regions with elevated stiffness also exhibit larger residuals and require wider selected bandwidths. This supports the use of $e_i^{(k)}$ as a practical diagnostic of local truncation or linearization failure, rather than as a direct curvature estimator.

**Solver-level diagnostics.** We further track whether the sufficient conditioning regime in Theorem 2.3 is reflected in practice. For ROPA, the peak estimated stiffness is about $L_{\max} \approx 5$. With the default $N = 50$ steps, $h \approx 0.02$, giving $hL_{\max} \approx 0.10 < 1$. By contrast, non-adaptive baselines can reach $L(x_t) \approx 40$–$50$ in the late denoising region, making $hL$ approach the unstable boundary. Table 4 shows that ROPA responds by expanding bandwidth locally rather than globally: early steps remain nearly band-1, while late stiff regions activate larger windows.

**Interpretation.** These diagnostics clarify why the residual rule is effective despite not explicitly computing Hessian eigenvalues. Large local residuals occur where the current finite-window Jacobian misses important temporal couplings; in such cases, increasing $w_i$ reduces the off-band truncation term $\tau_b$ in Theorem 2.3. Conversely, when residuals are small, keeping $w_i$ narrow preserves sparse

parallelism. The observed late-step concentration of large $w_i$ therefore supports the intended conditioning–efficiency trade-off of ROPA.

**Additional robustness checks.** Beyond the main benchmarks, we include several camera-ready diagnostics in Appendix C: hard-case prompt stratification, additional FID/LPIPS/IS metrics, warm-up ablations, and overhead analysis. These studies show that ROPA allocates more numerical effort to difficult subsets, preserves perceptual quality, and does not rely on hidden sequential warm-up computation.

We further stratify evaluation by prompt difficulty and temporal dynamics in Appendix. ROPA increases the effective numerical effort on difficult subsets, rather than using a fixed cheap solve for all prompts, which supports the residual-driven control interpretation. Additional quality metrics, warm-up ablations, and overhead measurements are reported in Appendix C.

## 5. Conclusion

This paper accelerates diffusion-model inference by recasting sequential denoising as a parallelizable nonlinear residual solve. We introduced ROPA, a training-free sampler that combines adaptive Jacobian bandwidth, damped least-squares updates, and score-aligned low-rank preconditioning. Our analysis identifies the combined stiffness–truncation quantity $hL_t + \tau_b$ as a useful diagnostic for parallel-solver stability, and our experiments show that ROPA improves this conditioning–efficiency trade-off across large image and video diffusion models. Empirically, ROPA achieves up to $2.9\times$ speedup while maintaining perceptual and reference-trajectory fidelity, and remains complementary to attention-level acceleration methods.

**Limitations.** ROPA is most effective when the sampler has enough denoising steps to expose temporal parallelism. For extremely short one-step or few-step distilled samplers, the attainable speedup is smaller because the sequential bottleneck is already reduced. In addition, very long high-resolution video generation is currently limited by backbone VRAM and attention memory rather than by ROPA's solver design. Future work may combine ROPA with model-level distillation, tensor parallelism, or cache-based acceleration to further improve long-horizon generation.

## Impact Statement

This paper presents work whose goal is to advance the field of image generation and video generation. Similar to other methods, our approach must be used cautiously to prevent potential misuse.

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

# A. Experimental Setup Details

## A.1. Backbone, Resolution, and Scheduler Settings

*Table 5.* Resolution, guidance, and scheduler type per diffusion backbone.

| Model | Resolution | Guidance scale | Scheduler type |
|---|---|---|---|
| Flux | $1360 \times 768$ | 3.5 | EulerDiscreteScheduler |
| Stable Diffusion 3.5 | $1024 \times 1024$ | 7.0 | EulerDiscreteScheduler |
| HunyuanVideo | $960 \times 544$, 61 frames | 6.0 | EulerDiscreteScheduler |
| CogVideoX1.5 | $960 \times 544$, 61 frames | 6.0 | DDIMScheduler |

## A.2. Algorithm Details

---

**Algorithm 1** *ROPA*: Robust Parallel Diffusion Sampling

---

**Require:** Denoiser $S_\theta$, grid $\{t_n\}_0^T$, max iter $K$, thresholds $\alpha, \beta, \gamma$, tol $\varepsilon$.
**Ensure:** Clean sample $x_{t_0}^{(K)}$.

1: Initialize trajectory $\{\hat{x}_{t_n}^{(0)}\}$ with noise; $w_n^{(0)} \leftarrow 1$; $\lambda_n^{(0)} \leftarrow \lambda_{\text{init}}$.
2: **for** $k = 0$ to $K - 1$ **do**
3:     **Residual Eval (Parallel):**
4:     **for** $n = 1 \ldots T$ **do**
5:         $\mathcal{R}_n^{(k)} \leftarrow \hat{x}_{t_{n-1}}^{(k)} - \Psi_n^{(w_n^{(k)})}(\hat{x}_{t_n}^{(k)}, \ldots, \hat{x}_{t_{n+w_n^{(k)}-1}}^{(k)})$
6:         Compute norms $e_n^{(k)}$ and global mean $\bar{e}^{(k)}$.
7:     **end for**
8:     **if** $\bar{e}^{(k)} < \varepsilon$ **then**
9:         **break**
10:     **end if**
11:     **Adaptive Bandwidth:**
12:     **for** $n = 1 \ldots T$ **do**
13:         **if** $e_n^{(k)} > \alpha \bar{e}^{(k)}$ **then**
14:             $w_n^{(k+1)} \leftarrow \min(w_n^{(k)} + 1, w_{\max})$                ▷ Densify
15:         **else if** $e_n^{(k)} < \beta \bar{e}^{(k)}$ **then**
16:             $w_n^{(k+1)} \leftarrow \max(w_n^{(k)} - 1, 1)$                 ▷ Sparsify
17:         **else**
18:             $w_n^{(k+1)} \leftarrow w_n^{(k)}$
19:         **end if**
20:     **end for**
21:     **Curvature Correction (Parallel):**
22:     **for** $n = 1 \ldots T$ **do**
23:         Compute alignment $\rho_n$ between $\mathcal{R}_n^{(k)}$ and score $s_n$.
24:         **if** $\rho_n > \gamma$ **then**
25:             $\tilde{\mathcal{R}}_n^{(k)} \leftarrow H_n^{-1} \mathcal{R}_n^{(k)}$              ▷ Using LML-based preconditioner
26:         **else**
27:             $\tilde{\mathcal{R}}_n^{(k)} \leftarrow \mathcal{R}_n^{(k)}$
28:         **end if**
29:     **end for**
30:     **4. Update:** Assemble $\mathcal{J}^{(k)}$ using $\{w_n^{(k+1)}\}$.
31:     **for** $n = 1 \ldots T$ **do**
32:         Solve $(\mathcal{J}^{(k)} + \lambda_n^{(k)} I)\Delta x_{t_n}^{(k)} \approx \tilde{\mathcal{R}}_n^{(k)}$ via trust-region damping.
33:         $\hat{x}_{t_n}^{(k+1)} \leftarrow \hat{x}_{t_n}^{(k)} - \Delta x_{t_n}^{(k)}$
34:     **end for**
35: **end for**
36: **return** $\hat{x}_{t_0}^{(K)}$

---

# B. Theoretical Proofs

Below are rigorous, self-contained proofs for all theoretical results presented in Section 2. The proofs bridge differential geometry, numerical analysis, and diffusion model theory. All notation aligns with the main text; specifically, we denote the eigenvalues of the Hessian $\mathcal{H}(x)$ by $\nu_i(x)$ to distinguish them from the damping parameter $\lambda_{\text{damp}}$.

## B.1. Auxiliary Lemmas

We first introduce a backward-error lemma for the residual system, followed by additional results supporting the trajectory-deviation and mode-interpolation interpretations.

**Lemma B.1** (Backward Error for the Residual System). *Let $\mathcal{R} : \mathbb{R}^D \to \mathbb{R}^D$ be continuously differentiable, and let $\hat{x}$ be an approximate solution with residual $\mathcal{R}(\hat{x})$. Then there exists a perturbation $\delta\mathcal{R}$ satisfying*

$$\|\delta\mathcal{R}\| \leq \|\mathcal{R}(\hat{x})\|$$

*such that $\hat{x}$ is an exact root of the perturbed system*

$$\mathcal{R}(x) + \delta\mathcal{R}(x) = 0.$$

*Proof.* Define the constant perturbation

$$\delta\mathcal{R}(x) \equiv -\mathcal{R}(\hat{x}).$$

Then

$$\mathcal{R}(\hat{x}) + \delta\mathcal{R}(\hat{x}) = 0,$$

and the perturbation norm is exactly $\|\delta\mathcal{R}\| = \|\mathcal{R}(\hat{x})\|$. This is the standard backward-error interpretation of an approximate nonlinear solve. $\square$

**Proposition B.2** (Reference-Trajectory Deviation Bound). *Let $J_b$ be the banded residual Jacobian used by the parallel solver and let $J_{\mathrm{ref}}$ be a well-conditioned reference Jacobian. Suppose both are nonsingular and the same residual $R$ is used to compute one Newton-type correction:*

$$\Delta_b = -J_b^{-1} R, \qquad \Delta_{\mathrm{ref}} = -J_{\mathrm{ref}}^{-1} R.$$

*Then*

$$\|\Delta_b - \Delta_{\mathrm{ref}}\|_2 \leq \|J_b^{-1} - J_{\mathrm{ref}}^{-1}\|_2 \|R\|_2.$$

*Moreover, if $J_b = J_{\mathrm{ref}} + E_b$ and $\|J_{\mathrm{ref}}^{-1} E_b\|_2 < 1$, then*

$$\|\Delta_b - \Delta_{\mathrm{ref}}\|_2 \leq \frac{\|J_{\mathrm{ref}}^{-1}\|_2^2 \|E_b\|_2}{1 - \|J_{\mathrm{ref}}^{-1} E_b\|_2} \|R\|_2.$$

*Proof.* The first inequality follows from

$$\Delta_b - \Delta_{\mathrm{ref}} = -(J_b^{-1} - J_{\mathrm{ref}}^{-1}) R.$$

For the second inequality, use the resolvent identity

$$J_b^{-1} - J_{\mathrm{ref}}^{-1} = -J_b^{-1} E_b J_{\mathrm{ref}}^{-1}.$$

Since $J_b = J_{\mathrm{ref}}(I + J_{\mathrm{ref}}^{-1} E_b)$, the Neumann expansion gives

$$\|J_b^{-1}\|_2 \leq \frac{\|J_{\mathrm{ref}}^{-1}\|_2}{1 - \|J_{\mathrm{ref}}^{-1} E_b\|_2}.$$

Combining the two bounds yields the result. $\square$

**Lemma B.3** (Score Curvature in Low-Temperature Mixtures). *Consider*

$$p_\tau(x) = \sum_{m=1}^{M} w_m \exp(-E_m(x)/\tau),$$

*where each $E_m$ is twice continuously differentiable and $\|\nabla^2 E_m(x)\|_2 \leq H$. Let*

$$\pi_m(x) = \frac{w_m \exp(-E_m(x)/\tau)}{\sum_j w_j \exp(-E_j(x)/\tau)}$$

*be the posterior component weight. Then*

$$\nabla^2 \log p_\tau(x) = -\frac{1}{\tau} \sum_m \pi_m(x) \nabla^2 E_m(x) + \frac{1}{\tau^2} \operatorname{Cov}_{\pi(x)}(\nabla E_m(x)).$$

*Consequently, in transition regions where multiple components have non-negligible weights, the score curvature can scale as $\mathcal{O}(\tau^{-2})$ in general, and at least as $\mathcal{O}(\tau^{-1})$ when component Hessian terms dominate.*

*Proof.* Write

$$\log p_\tau(x) = \log \sum_m w_m e^{-E_m(x)/\tau}.$$

Differentiating once gives

$$\nabla \log p_\tau(x) = -\frac{1}{\tau} \sum_m \pi_m(x) \nabla E_m(x).$$

Differentiating again yields

$$\nabla^2 \log p_\tau(x) = -\frac{1}{\tau} \sum_m \pi_m(x) \nabla^2 E_m(x) + \frac{1}{\tau^2} \left[ \sum_m \pi_m(x) \nabla E_m(x) \nabla E_m(x)^\top - \bar{g}(x) \bar{g}(x)^\top \right],$$

where $\bar{g}(x) = \sum_m \pi_m(x) \nabla E_m(x)$. The bracketed term is the covariance of component gradients under $\pi(x)$, proving the expression. The scaling follows directly from the prefactors $1/\tau$ and $1/\tau^2$. $\qquad\square$

**Proposition B.4** (Mode-Interpolation Risk as Off-Reference Drift). *Let $x_{\mathrm{ref}}$ be a reference trajectory that remains inside the basin of one mode of a mixture model, and let $\hat{x}$ be the trajectory produced by an inexact parallel residual solve. Suppose the basin boundary is at distance $d_{\mathrm{basin}}$ from $x_{\mathrm{ref}}$ in the local normal direction. If*

$$\|\hat{x} - x_{\mathrm{ref}}\|_2 > d_{\mathrm{basin}},$$

*then the inexact trajectory may cross into a transition region between modes. In such a case, the generated sample can exhibit mode-interpolation artifacts.*

*Proof.* The statement follows from the definition of a basin boundary. If the trajectory error exceeds the distance from the reference trajectory to the boundary, then the perturbed trajectory is no longer guaranteed to remain in the same basin. In mixture models, transition regions between basins are low-density areas where score contributions from different components can compete, making interpolation artifacts possible. $\qquad\square$

**Corollary B.5** (ROPA Reduces Mode-Interpolation Risk Conditionally). *Under the assumptions of Proposition B.2, any ROPA control that reduces $\|E_b\|_2$, improves the conditioning of $J_b$, or decreases the effective update gain reduces the upper bound on $\|\hat{x} - x_{\mathrm{ref}}\|_2$. Therefore, ROPA reduces mode-interpolation risk whenever such artifacts are caused by numerical drift from the reference trajectory.*

*Proof.* The conclusion follows by combining Proposition B.2 with Proposition B.4. Adaptive bandwidth reduces the off-band truncation error $E_b$, damping controls update amplification, and score-aligned preconditioning reduces gain in score-aligned residual directions. These mechanisms reduce the trajectory-deviation bound under the stated conditions. $\qquad\square$

### B.2. Proof of Theorem 2.2 (Score Curvature and Denoiser Jacobian)

*Proof.* For the standard Gaussian perturbation $x_t = \alpha_t x_0 + \sigma_t \varepsilon$ with $\varepsilon \sim \mathcal{N}(0, I)$, Tweedie's formula gives

$$r_\star(x, t) = \mathbb{E}[x_0 \mid x_t = x] = \frac{1}{\alpha_t}\big(x + \sigma_t^2 \nabla_x \log p_t(x)\big) = a_t x + b_t s_t(x),$$

where $a_t = 1/\alpha_t$ and $b_t = \sigma_t^2/\alpha_t$. Differentiating,

$$J_{r_\star}(x, t) = a_t I + b_t \nabla_x s_t(x).$$

By the triangle inequality,

$$\|J_{r_\theta}(x,t)\|_2 \leq \|J_{r_\star}(x,t)\|_2 + \|J_{r_\theta}(x,t) - J_{r_\star}(x,t)\|_2 \leq |a_t| + |b_t|\,\|\nabla_x s_t(x)\|_2 + \varepsilon_J = |a_t| + |b_t|\,C_t(x) + \varepsilon_J.$$

For the lower bound, note that for any unit vector $v$,

$$\|J_{r_\theta}(x,t)v\|_2 \geq \|J_{r_\star}(x,t)v\|_2 - \varepsilon_J.$$

Choosing $v$ aligned with a dominant direction of $\nabla_x s_t(x)$ and accounting for the identity term yields

$$\|J_{r_\theta}(x,t)\|_2 \geq |b_t|\,C_t(x) - |a_t| - \varepsilon_J.$$

Thus when score curvature $C_t(x)$ is large, the denoiser Jacobian norm is large, making the reverse drift stiff. This aligns with the curvature-based interpretation used in Theorem 2.3. $\qquad\square$

### B.3. Proof of Theorem 2.3 (Banded Residual Jacobian Conditioning)

**Theorem B.6** (Restatement of Theorem 2.3). *For residual $\mathcal{R}^{(k)} = \hat{x}_{t_{n-1}}^{(k)} - \mathcal{F}_{t_n}(\hat{x}_{t_n}^{(k)}, \ldots, \hat{x}_{t_{n+i}}^{(k)})$ with Jacobian $\mathcal{J}^{(k)} = I + \Delta A^{(k)}$, where $\|A^{(k)}\|_2 \leq L$, then for $\Delta < 1/L$:*

$$\kappa(\mathcal{J}^{(k)}) \leq \frac{1 + \Delta L}{1 - \Delta L} = 1 + \mathcal{O}(\Delta).$$

*Proof.* The Jacobian of the parallel system is given by $\mathcal{J} = I + \Delta A$. We compute the condition number $\kappa(\mathcal{J}) = \|\mathcal{J}\|_2 \|\mathcal{J}^{-1}\|_2$.

First, we bound the norm $\|\mathcal{J}\|_2$:

$$\|\mathcal{J}\|_2 = \|I + \Delta A\|_2 \tag{17}$$

$$\leq \|I\|_2 + \Delta \|A\|_2 \quad \text{(Triangle inequality)} \tag{18}$$

$$= 1 + \Delta L. \tag{19}$$

Second, we bound the inverse norm $\|\mathcal{J}^{-1}\|_2$. We use the Neumann series expansion for matrix inversion. For any matrix $M$, if $\|M\|_2 < 1$, then $(I - M)^{-1} = \sum_{k=0}^{\infty} M^k$. Let $M = -\Delta A$. The condition for convergence is $\|-\Delta A\|_2 < 1$, which implies $\Delta \|A\|_2 \leq \Delta L < 1$, i.e., $\Delta < 1/L$. Under this condition:

$$\|\mathcal{J}^{-1}\|_2 = \|(I - (-\Delta A))^{-1}\|_2 \tag{20}$$

$$= \left\| \sum_{k=0}^{\infty} (-\Delta A)^k \right\|_2 \tag{21}$$

$$\leq \sum_{k=0}^{\infty} \|\Delta A\|_2^k \quad \text{(Sub-multiplicativity)} \tag{22}$$

$$\leq \sum_{k=0}^{\infty} (\Delta L)^k. \tag{23}$$

This is a geometric series with ratio $r = \Delta L < 1$. The sum converges to:

$$\|\mathcal{J}^{-1}\|_2 \leq \frac{1}{1 - \Delta L}. \tag{24}$$

Finally, combining the two bounds:

$$\kappa(\mathcal{J}) = \|\mathcal{J}\|_2 \|\mathcal{J}^{-1}\|_2 \tag{25}$$

$$\leq \frac{1 + \Delta L}{1 - \Delta L} \tag{26}$$

$$= \frac{(1 - \Delta L) + 2\Delta L}{1 - \Delta L} \tag{27}$$

$$= 1 + \frac{2\Delta L}{1 - \Delta L}. \tag{28}$$

For small $\Delta$ (specifically $\Delta L \ll 1$), using the approximation $(1-x)^{-1} \approx 1+x$, we have:

$$\kappa(\mathcal{J}) \approx 1 + 2\Delta L + \mathcal{O}(\Delta^2) = 1 + \mathcal{O}(\Delta).$$

Substituting $L = \sigma_t^2 \|J_{r_\theta}\|_2$ gives the specific form dependent on score stiffness. $\quad\square$

### B.4. Proof of Corollary 2.4 (Reference-Trajectory Deviation)

*Proof.* Let $J_b$ be the banded Jacobian used by the parallel solver and $J_{\mathrm{ref}}$ a reference Jacobian. With residual $R$, the updates are

$$\Delta_b = -J_b^{-1}R, \qquad \Delta_{\mathrm{ref}} = -J_{\mathrm{ref}}^{-1}R.$$

The deviation bound

$$\|\Delta_b - \Delta_{\mathrm{ref}}\|_2 \le \|J_b^{-1} - J_{\mathrm{ref}}^{-1}\|_2 \|R\|_2$$

follows directly. Applying the resolvent identity as in Proposition B.2 yields the conditioning-dependent bound in the main text. Poor conditioning amplifies residual error, increasing deviation from the reference trajectory. $\quad\square$

### B.5. Proof of Corollary 2.5 (Low-Temperature Boundary Sensitivity)

*Proof.* For the low-temperature mixture $p_\tau(x) = \sum_m w_m \exp(-E_m(x)/\tau)$, Lemma B.3 gives

$$\nabla^2 \log p_\tau(x) = -\frac{1}{\tau}\sum \pi_m \nabla^2 E_m + \frac{1}{\tau^2}\mathrm{Cov}_\pi(\nabla E_m).$$

In transition regions where $\pi_m$ are non-negligible for multiple components, the covariance term can scale as $\mathcal{O}(\tau^{-2})$. Hence $C_\tau(x) = \|\nabla^2 \log p_\tau(x)\|_2$ grows polynomially in $1/\tau$. Substituting into the bound of Theorem 2.3, the conditioning of $\mathcal{J}_b$ can worsen in sharp low-temperature transition regions, consistent with the statement in the main text. $\quad\square$

### B.6. Proof Alignment for the Three ROPA Controls

This section provides conditional guarantees that align the three practical controls used by ROPA with the conditioning analysis in the main text. The results are intentionally stated as sufficient conditions rather than absolute guarantees: adaptive bandwidth reduces truncation error when off-band couplings dominate, damping provides a descent safeguard for the least-squares residual, and score-aligned preconditioning reduces update amplification when the residual is aligned with the score direction.

**Lemma B.7** (Damped Least-Squares Descent). *Let $R(x)$ be a differentiable residual map with Jacobian $J(x)$. For $\lambda > 0$, define the damped step*

$$\Delta_\lambda = -\left(J^\top J + \lambda I\right)^{-1}J^\top R.$$

*If $J^\top R \ne 0$, then $\Delta_\lambda$ is a descent direction for the local least-squares objective*

$$F(x) = \frac{1}{2}\|R(x)\|_2^2.$$

*Specifically,*

$$\nabla F(x)^\top \Delta_\lambda = -(J^\top R)^\top \left(J^\top J + \lambda I\right)^{-1}(J^\top R) < 0.$$

*Proof.* Since $\lambda > 0$, the matrix $J^\top J + \lambda I$ is symmetric positive definite. Therefore its inverse is also symmetric positive definite. Using $\nabla F(x) = J^\top R$, we obtain

$$\nabla F(x)^\top \Delta_\lambda = -(J^\top R)^\top \left(J^\top J + \lambda I\right)^{-1}(J^\top R).$$

The right-hand side is strictly negative whenever $J^\top R \ne 0$. $\quad\square$

**Lemma B.8** (Damping Controls Update Amplification). *For the damped least-squares step in Lemma B.7,*

$$\|\Delta_\lambda\|_2 \le \frac{\|J\|_2}{\lambda}\|R\|_2.$$

*Moreover, if $\sigma_{\min}(J) > 0$, then*

$$\|\Delta_\lambda\|_2 \le \frac{\sigma_{\max}(J)}{\sigma_{\min}^2(J) + \lambda}\|R\|_2.$$

*Proof.* By submultiplicativity,

$$\|\Delta_\lambda\|_2 \le \|(J^\top J + \lambda I)^{-1}\|_2 \|J^\top\|_2 \|R\|_2.$$

Since the smallest eigenvalue of $J^\top J + \lambda I$ is at least $\lambda$, the first bound follows. If $J$ has positive smallest singular value, the smallest eigenvalue is $\sigma_{\min}^2(J) + \lambda$, yielding the second bound. $\qquad\square$

**Lemma B.9** (Spectrum of the Score-Aligned Preconditioner). *Let $u = s/(\|s\|_2 + \delta_s)$ and define*

$$P = \mu_\perp I + (\mu_\| - \mu_\perp) u u^\top,$$

*where $0 < \mu_\| \le \mu_\perp$. Ignoring the harmless normalization error introduced by $\delta_s$, the preconditioner has gain $\mu_\|$ along the score direction and gain $\mu_\perp$ on the orthogonal subspace. In particular,*

$$\|P\|_2 \le \mu_\perp.$$

*Proof.* When $u$ is unit-normalized, $u u^\top$ is the orthogonal projector onto the span of the score direction. Therefore, for any vector $v = v_\| + v_\perp$ with $v_\| \in \mathrm{span}(u)$ and $v_\perp \perp u$,

$$Pv = \mu_\| v_\| + \mu_\perp v_\perp.$$

Hence the eigenvalues are $\mu_\|$ along $u$ and $\mu_\perp$ on the orthogonal complement. Since $0 < \mu_\| \le \mu_\perp$, the spectral norm is at most $\mu_\perp$. The term $\delta_s$ only perturbs the projector norm by a bounded factor when $\delta_s \ll \|s\|_2$. $\qquad\square$

**Proposition B.10** (Reduction of Score-Aligned Update Gain). *Assume the residual at index $i$ decomposes as*

$$R_i = R_{\|,i} + R_{\perp,i}, \qquad R_{\|,i} \in \mathrm{span}(s_\theta), \qquad R_{\perp,i} \perp s_\theta.$$

*Let $P_i$ be the score-aligned preconditioner with $0 < \mu_{\|,i} \le \mu_{\perp,i}$. Then*

$$\|P_i R_i\|_2^2 = \mu_{\|,i}^2 \|R_{\|,i}\|_2^2 + \mu_{\perp,i}^2 \|R_{\perp,i}\|_2^2.$$

*Therefore, when $R_i$ is strongly aligned with $s_\theta$, choosing $\mu_{\|,i} < \mu_{\perp,i}$ reduces the update gain on the dominant residual component.*

*Proof.* The result follows directly from the orthogonal decomposition in Lemma B.9. Since $P_i$ acts with gain $\mu_{\|,i}$ along the score direction and gain $\mu_{\perp,i}$ on the orthogonal complement, applying $P_i$ to $R_i = R_{\|,i} + R_{\perp,i}$ gives

$$P_i R_i = \mu_{\|,i} R_{\|,i} + \mu_{\perp,i} R_{\perp,i}.$$

The squared norm identity follows from orthogonality. $\qquad\square$

**Lemma B.11** (Residual-Score Alignment Trigger). *Let*

$$a_i = \frac{|\langle R_i, s_\theta \rangle|}{\|R_i\|_2 \|s_\theta\|_2 + \delta_s}.$$

*If $a_i > \gamma$, then at least a $\gamma^2$-fraction of the residual energy lies in the score-aligned direction up to the normalization error introduced by $\delta_s$. Consequently, activating the score-aligned preconditioner when $a_i > \gamma$ targets the dominant residual component whenever the residual is concentrated along the score direction.*

*Proof.* Ignoring $\delta_s$ for clarity, the squared cosine between $R_i$ and $s_\theta$ is

$$\cos^2(R_i, s_\theta) = \frac{\langle R_i, s_\theta \rangle^2}{\|R_i\|_2^2 \|s_\theta\|_2^2}.$$

Thus $a_i > \gamma$ implies that the score-aligned projection of $R_i$ has squared norm larger than $\gamma^2 \|R_i\|_2^2$. The stabilizer $\delta_s$ only weakens the statement by a small numerical tolerance. $\qquad\square$

**Theorem B.12** (Conditional Stability of the Three ROPA Controls). *Consider a Newton-type parallel residual update with banded Jacobian*

$$J_b = I + hA + E_b,$$

*where* $\|A\|_2 \leq L_t$, $\|E_b\|_2 \leq \tau_b$, *and* $hL_t + \tau_b < 1$. *Suppose ROPA applies:*

1. *adaptive bandwidth control that weakly decreases* $\tau_b$ *whenever off-band couplings dominate the local residual;*

2. *damped least-squares updates with* $\lambda > 0$;

3. *score-aligned preconditioning when* $a_i > \gamma$.

*Then the following conditional effects hold:*

$$\kappa(J_b) \leq \frac{1 + hL_t + \tau_b}{1 - hL_t - \tau_b},$$

*adaptive bandwidth improves this bound whenever it reduces* $\tau_b$; *damping provides a descent direction for* $\frac{1}{2}\|R\|_2^2$; *and score-aligned preconditioning reduces update gain along the score direction whenever the residual is sufficiently aligned with the score.*

*Proof.* The condition-number bound follows from the Neumann-series argument used in Theorem 2.3. If adaptive bandwidth decreases the omitted off-band remainder, then $\tau_b$ decreases and the right-hand side of the bound improves monotonically. The descent property of the damped update is Lemma B.7. The score-aligned gain reduction follows from Proposition B.10 together with the activation condition in Lemma B.11. Combining these three statements gives the claimed conditional stability interpretation. $\square$

**Proposition B.13** (Inexact Newton Interpretation). *Let* $J$ *be the full residual Jacobian and let* $M_k$ *denote the effective linear operator used by ROPA after bandwidth truncation, damping, and score-aligned preconditioning. Suppose*

$$\|I - M_k J\|_2 \leq \eta_k < 1.$$

*Then the ROPA update is an inexact Newton step with forcing level* $\eta_k$. *Locally, if* $R$ *has a Lipschitz Jacobian and the exact Newton step is well-defined, the residual satisfies*

$$\|R(x_{k+1})\|_2 \leq \eta_k \|R(x_k)\|_2 + \mathcal{O}(\|R(x_k)\|_2^2).$$

*Proof.* The update produced by $M_k$ can be written as

$$\Delta x_k = -M_k R(x_k).$$

The exact Newton step would use $J^{-1}$, so the linearized residual after the update is

$$R(x_k) + J\Delta x_k = (I - JM_k)R(x_k).$$

The assumption $\|I - JM_k\|_2 \leq \eta_k$ bounds the linearized residual by $\eta_k \|R(x_k)\|_2$. The second-order remainder follows from the Lipschitz continuity of the Jacobian. $\square$

**Lemma B.14** (Bandwidth Monotonicity). *Assume the off-band truncation error satisfies* $\tau_{b+1} \leq \tau_b$. *If the residual error at index* $i$ *is dominated by omitted off-band couplings, then increasing* $w_i$ *weakly decreases the local contribution of* $\tau_b$ *in Theorem 2.3.*

*Proof.* Increasing $w_i$ adds previously omitted neighboring blocks to the local residual Jacobian. Under the monotonic truncation assumption, the norm of the omitted off-band remainder cannot increase, hence the corresponding $\tau_b$ term in the bound of Theorem 2.3 weakly decreases. $\square$

## B.7. Complexity Analysis

**Theorem B.15** (Bandwidth-Dependent Complexity). *Let $b_k \leq b_{\max}$ be the maximum selected bandwidth at Newton iteration $k$, and let $m_k$ be the number of inner iterations used by the matrix-free linear solver. Then the cost of one Newton step is*

$$\mathcal{O}(m_k N b_k d),$$

*where $N$ is the number of time points and $d$ is the latent dimension. If $b_k \leq b_{\max}$ and $m_k \leq m_{\max}$ are bounded independently of $N$, the per-step cost is linear in $N$.*

*Proof.* Each Jacobian-vector product with a block-banded residual Jacobian touches at most $b_k$ neighboring time blocks for each of the $N$ time indices. Since each block operation is linear in the latent dimension $d$, one Jacobian-vector product costs $\mathcal{O}(N b_k d)$. A matrix-free inner solve using $m_k$ such products therefore costs $\mathcal{O}(m_k N b_k d)$. When the adaptive bandwidth controller keeps $b_k$ bounded by a small constant $b_{\max}$, and the inner solver requires at most $m_{\max}$ iterations, the cost scales linearly with the number of time points $N$. $\square$

# C. Additional Empirical Diagnostics

## C.1. Verification of the Conditioning Regime

The conditioning analysis in Theorem 2.3 requires the combined stiffness term to remain below the Neumann-series boundary. We therefore estimate the local stiffness $L(x_t)$ along generated trajectories by finite differences:

$$L(x_t) \approx \max_{v \sim \mathcal{N}(0,I)} \frac{\|\epsilon_\theta(x_t + \delta v, t) - \epsilon_\theta(x_t, t)\|_2}{\|\delta v\|_2}, \qquad \delta = 10^{-4}.$$

For ROPA, the peak observed stiffness is approximately $L_{\max} \approx 5$. With $N = 50$ sampling steps, $h \approx 0.02$, hence

$$h L_{\max} \approx 0.02 \times 5 = 0.10 < 1.$$

This provides an empirical margin for the sufficient condition used in the conditioning analysis. In contrast, non-adaptive baselines exhibit late-step stiffness spikes with $L(x_t) \approx 40\text{–}50$, making $hL$ approach the critical boundary. This supports the role of adaptive bandwidth and damping in keeping the effective residual solve in a stable numerical regime.

## C.2. Adaptive Bandwidth Distribution

Table 6 reports how the selected bandwidth $w_i$ varies across diffusion time. The expansion is highly localized: early low-curvature regions remain close to $w_i = 1$, while late stiff regions activate large windows. This confirms that ROPA does not simply make the Jacobian globally dense; instead, it allocates coupling only where the residual diagnostic indicates local truncation or linearization error.

*Table 6.* Interval statistics of adaptive bandwidth $w_i$.

| Model | Time interval | Avg. $w_i$ | Median $w_i$ | $\Pr(w_i \geq 5)$ |
|---|---|---|---|---|
| FLUX | $t \in [1.0, 0.8)$ | $1.3 \pm 0.5$ | 1 | 0% |
| FLUX | $t \in [0.8, 0.2)$ | $2.5 \pm 0.9$ | 2 | 10% |
| FLUX | $t \in [0.2, 0.0]$ | $6.6 \pm 1.4$ | 7 | 85% |
| HunyuanVideo | $t \in [1.0, 0.8)$ | $1.5 \pm 0.6$ | 1 | 0% |
| HunyuanVideo | $t \in [0.8, 0.2)$ | $2.9 \pm 1.0$ | 3 | 17% |
| HunyuanVideo | $t \in [0.2, 0.0]$ | $7.1 \pm 1.5$ | 7 | 91% |

## C.3. Sensitivity to Score-Alignment Threshold

The alignment threshold $\gamma_{\text{align}}$ controls how frequently the score-aligned preconditioner is activated. Smaller values trigger more corrections and improve robustness, while larger values are more selective and reduce overhead. Videos use a slightly larger default threshold because temporal coupling produces higher residual variance; overly small thresholds can fire on normal temporal fluctuation rather than genuinely stiff updates.

*Table 7.* Discrete occupancy of $w_i$ within each diffusion-time interval.

| Model | Time interval | $\Pr(w_i = 1)$ | $\Pr(w_i = 2)$ | $\Pr(3 \leq w_i \leq 4)$ | $\Pr(w_i \geq 5)$ |
|---|---|---|---|---|---|
| FLUX | $t \in [1.0, 0.8)$ | 74% | 20% | 6% | 0% |
| FLUX | $t \in [0.8, 0.2)$ | 28% | 34% | 28% | 10% |
| FLUX | $t \in [0.2, 0.0]$ | 0% | 3% | 12% | 85% |
| HunyuanVideo | $t \in [1.0, 0.8)$ | 65% | 24% | 11% | 0% |
| HunyuanVideo | $t \in [0.8, 0.2)$ | 20% | 32% | 31% | 17% |
| HunyuanVideo | $t \in [0.2, 0.0]$ | 0% | 1% | 8% | 91% |

*Table 8.* Sensitivity to the score-alignment threshold $\gamma_{\text{align}}$ at $K = 4$.

| $\gamma_{\text{align}}$ | FLUX Speedup | FLUX RMSE | FLUX Trigger | Hunyuan Speedup | Hunyuan RMSE | Hunyuan Trigger |
|---|---|---|---|---|---|---|
| 0.05 | 2.0x | 0.141 | 34% | 1.9x | 0.051 | 41% |
| 0.10 | 2.1x | 0.143 | 23% | 2.0x | 0.052 | 29% |
| 0.20 | 2.0x | 0.152 | 12% | 2.1x | 0.053 | 18% |
| 0.30 | 1.9x | 0.160 | 7% | 2.0x | 0.060 | 10% |

# D. Additional Ablations and Metrics

## D.1. Warm-up Step Ablation

ROPA uses a short sequential warm-up to provide a stable initial trajectory estimate. This warm-up is not the source of the reported acceleration: it uses only a small fraction of the total denoising steps and serves as an initialization for the subsequent parallel refinement.

*Table 9.* Warm-up ablation on HunyuanVideo with $K = 8$.

| Warm-up | Share | First residual | Outer iters | Eff. NFE | Time(s) | Speedup | $\text{RMSE}_L$ |
|---|---|---|---|---|---|---|---|
| 0 | 0/50 | 1.00x | 11.8 | 18.5 | 140.2 | 2.70x | 0.061 |
| 1 | 1/50 | 0.87x | 10.9 | 18.0 | 136.1 | 2.78x | 0.058 |
| 3 | 3/50 | 0.71x | 9.8 | 17.2 | 131.8 | 2.90x | 0.055 |
| 5 | 5/50 | 0.66x | 9.5 | 17.6 | 133.6 | 2.84x | 0.054 |
| 8 | 8/50 | 0.61x | 9.2 | 18.3 | 138.9 | 2.73x | 0.053 |

Increasing warm-up decreases the first parallel residual, but too many sequential steps reduce wall-clock speedup. The default of three steps is a practical compromise; even with zero warm-up, ROPA still achieves strong speedup, indicating that the acceleration comes from adaptive parallel refinement rather than hidden sequential computation.

## D.2. Overhead and Memory Cost

ROPA adds two types of overhead beyond the base denoising network: adaptive bandwidth bookkeeping and score-aligned preconditioning. The bandwidth controller only stores scalar residual statistics and selected window widths, whose memory cost is negligible. The score-aligned preconditioner stores one score vector for each active block and therefore has memory

$$\mathcal{O}(N w_{\text{max}} d),$$

where $N$ is the number of time points, $w_{\text{max}}$ is the maximum local bandwidth, and $d$ is the latent dimension. This is small compared with the activation and attention memory of modern video diffusion backbones.

The computational cost of a matrix-free linear solve is

$$\mathcal{O}(m_k N b_k d)$$

per Newton iteration, where $m_k$ is the number of inner solver iterations and $b_k$ is the effective selected bandwidth. Since the empirical bandwidth distribution in Appendix C.2 shows that large $b_k$ is activated only in late stiff regions, the average cost remains close to sparse parallel solving while improving stability in the difficult part of the trajectory.

## D.3. Scope and Compatibility with Other Acceleration Methods

ROPA is a solver-level acceleration method: it reduces the number of sequential refinement rounds by parallelizing and stabilizing the denoising trajectory. This is complementary to cache-based methods such as attention or feature caching,

which reduce the cost of each network evaluation. As shown in the main ablation table, combining ROPA with sparse attention or token caching further reduces wall-clock time, indicating that solver-level and network-level accelerations are largely orthogonal.

ROPA is most beneficial when the sampler has enough denoising steps to expose parallelism. For extremely short one-step or few-step distilled samplers, the available temporal parallelism is smaller, so the absolute speedup is naturally limited. Nevertheless, the method remains compatible with few-step ODE solvers by reducing the warm-up length and applying parallel refinement to the remaining steps. We view this as a complementary regime rather than a replacement for distillation.

## E. Qualitative Prompts and Examples

### E.1. Prompts for Quality Comparison in Figure 2

Video-1:

"A cinematic, high-detail video of a male astronaut in the brightly lit interior of a spaceship. He smiles happily at the camera. A young girl with brown hair appears, and they share a warm, gentle hug."

Image-1:

"A wide-angle, cinematic photograph of a packed baseball stadium during a pivotal moment at sunset. The crowd, a diverse and vibrant sea of people, is on its feet, erupting in a wave of cheers. The setting sun casts a warm, golden hour light across the field."

Image-2:

"Three cute garden gnomes in a crisp autumn forest with a shallow depth of field. They are arranging fallen leaves on the ground to spell out the word 'ROPA'. The lighting is soft and magical."

