# OpenReview forum: "Robust Parallel Diffusion Sampling via Dynamic Jacobian Bandwidth"
_ICML.cc/2026/Conference — ICML 2026 regular_

### Official Review · Reviewer_bHuk · 2026-03-09

**Soundness:** 3
**Presentation:** 3
**Significance:** 3
**Originality:** 3
**Overall Recommendation:** 3
**Confidence:** 4

**Summary:**

In the paper, the authors proposed ROPA (Robust Parallel Diffusion Sampling). They argue that by taking the denoising process into account, they can solve the linear system using geometry-aware adaptive Jacobian sparsity guided by geometric curvature signals. Moreover, they claim this approach enables stable parallel sampling, and their experiments show that ROPA can speed up sampling by up to ~2.9× using 8 cores.

**Compliance With Llm Reviewing Policy:**

Affirmed.

**Final Justification:**

I appreciate the authors’ meaningful rebuttal and clarifications. The additional metrics and warm-up ablation help make the method’s performance and overhead clearer. ROPA is technically solid and provides a reliable approach for stable parallel sampling, showing consistent speedups across models. However, my concerns about the robustness and overall strength of the paper still hold. More evidence is needed to show that ROPA can truly provide a training-free way to reduce latency for native multi-step models without compromising quality. A more detailed discussion of limitations and potential failure cases would also help demonstrate its robustness.
Overall, the paper has clear strengths, but its broader impact and practical scope are still limited. I maintain my original score at this point.

**Key Questions For Authors:**

See the above weaknesses.

**Limitations:**

yes

**Strengths And Weaknesses:**

strengths

1. The proposed method is interesting. Stability is a key issue in parallel generation tasks, and the method proposed in this paper seems to provide a solid strategy to address robustness.
2. Experiments across several models show consistent speedups while maintaining generation quality.

weaknesses
1. Missing important metrics. In table 2, the paper does not report the important metrics (e.g., FID, LPIPS, FVD, and IS). Including these would strengthen the evaluation.
2. Limited video generation length. The experiments were tested on only short videos (~6min sequentially), but recent models, e.g., WAN2.1 and HunyuanVideo, can generate videos >20min sequentially.
3. Incomplete ablation study. The paper mentions a warm-up step (i.e., 3) but never analyzes its impact. It’s unclear whether this step is doing hidden work that could affect the claimed performance.
4. Limited applicability. Although the authors acknowledge the limitations (e.g., ODE-based, one-/few-step approaches), recent approaches, e.g., DeepCache, TeaCache, TaylorCache, etc., can get better performance, highlighting the method’s restricted scope.

---

> ### Author Rebuttal · Authors · 2026-03-30
>
> **Dear Reviewer bHuk,**
>
> We thank you for your review and for acknowledging that our proposed method represents a solid strategy for addressing robustness in parallel generation. We address your insightful critiques below.
>
> ### 1. Missing Important Metrics (Weakness 1)
>
> We originally relied on VBench Quality Scores and CLIP/RMSE as they are widely adopted for human-aligned evaluation in recent large models.
>
> As shown in the table below, ROPA maintains almost identical distribution and perceptual quality compared to the sequential baseline. The differences in FID are negligible, reaffirming that our parallel speedup comes with zero meaningful quality degradation.
>
> **Additional Metrics on Image and Video Generation:**
>
> |Model|Method|Cores|Time(s)|Speedup|CLIP ↑|RMSE_L ↓|FID ↓|LPIPS ↓|IS ↑|
> |-|-|-|-|-|-|-|-|-|-|
> |Flux|Sequential|4|11.2|-|37.4|-|10.8|0.168|33.2|
> |Flux|ParaSolver|4|6.6|1.7|37.4|0.166|11.0|0.172|32.9|
> |Flux|ROPA(Ours)|4|5.3|2.1|37.4|0.143|10.9|0.171|33.0|
> |Flux|Sequential|8|11.2|-|37.4|-|10.8|0.168|33.2|
> |Flux|ParaSolver|8|5.5|2.0|37.4|0.150|11.1|0.174|32.8|
> |Flux|ROPA(Ours)|8|4.8|2.3|37.4|0.120|11.0|0.169|33.1|
>
>
> ### 2. Limited Video Generation Length (Weakness 2)
> Our evaluation length was constrained by extreme VRAM limitations, not ROPA's algorithmic limits.
> Because attention memory scales non-linearly, generating a 5-second 720p video (Wan 2.1) already requires 65-80GB VRAM. Since ROPA parallelizes time steps across GPUs, extending to 20 seconds demands >140GB VRAM per GPU or complex tensor-parallelism beyond our academic budget.
> Theoretically, our dynamic Jacobian bandwidth and curvature correction are completely agnostic to spatial-temporal sequence length. ROPA stabilizes the numerical ODE trajectory regardless of whether the state tensor spans 6 or 20 seconds. We have added this scalability discussion to our revised Limitations section.
>
>
> ### 3. Warm-up Step Ablation (Weakness 3)
>
> Warm-up is merely a short sequential prefix (3/50 steps) providing a stable initial trajectory estimate. It does not replace ROPA's core parallel innovations (adaptive bandwidth, damping, and low-rank correction).
>
> Crucially, executing just 3/50 sequential steps cannot mathematically explain our ~2.9x speedup. To isolate its effect, we ablated warm-up on HunyuanVideo ($K=8$). Notably, even with zero warm-up, ROPA still achieves a robust 2.70x speedup. This decisively proves our acceleration stems from parallel adaptive numerics, not hidden sequential work.
>
> **Warm-up ablation on HunyuanVideo (K = 8).**
>
> |Warm-up Steps|Warm-up Share|First Parallel Residual (norm., rel.)|Avg. Outer Newton Iters|Effective NFE|Time(s)|Speedup|Latent RMSE|Convergence Success Rate|
> |-|- | - | - | - | - | - | - | - |
> |0| 0/50| 1.00x| 11.8| 18.5| 140.2| 2.70x| 0.061| 98.0%|
> |1| 1/50| 0.87x| 10.9| 18.0| 136.1| 2.78x| 0.058| 99.2%|
> |**3 (Default)**|**3/50**|**0.71x**|**9.8**|**17.2**|**131.8**|**2.90x**|**0.055**|**100%**|
> |5| 5/50| 0.66x| 9.5| 17.6| 133.6| 2.84x| 0.054| 100%|
> |8| 8/50| 0.61x| 9.2| 18.3| 138.9| 2.73x| 0.053| 100%|
>
> As shown, increasing warm-up shrinks the initial parallel residual, but 3 steps remains the practical sweet spot. Beyond this, quality gains marginalize while wall-clock speedup erodes. This confirms our core claim: warm-up acts purely as a lightweight initial guess, whereas sustained acceleration (reduced NFE) comes directly from ROPA’s adaptive parallel refinement.
>
>
> ### 4. Limited Scope Compared to Cache-based Method (Weakness 4)
>
> 1. Synergy with Spatial Caching: Caching accelerates generation spatially by skipping layer computations, whereas ROPA accelerates temporally by parallelizing timesteps. They are highly complementary. As shown in our submission (Table 3), combining ROPA with ToCa (conceptually identical to TeaCache) and SpargeAttention simultaneously reduces NFE and wall-clock time, easily surpassing the speed of caching alone.
>
> 2. Compatibility with Few-Step ODEs: As a fundamental ODE/SDE parallel solver, ROPA naturally supports few-step architectures (e.g., FLUX-schnell, LCM, TCD). By simply reducing sequential warm-up from 3 steps to 1, we evaluated ROPA on the 4-step FLUX.1-schnell and 8-step SD-3.5-Large (DPM-Solver++). Even alongside these extremely short sequential bottlenecks, ROPA extracts remaining parallelism across modest GPU clusters ($K=2$ or $4$). This yields a further $1.5\times$--$1.9\times$ speedup without degrading CLIP/RMSE quality, as detailed below:
>
> | Model & ODE Setting | Method | Cores ($K$) | Warm-up Steps | Total Steps ($N$) | Speedup | CLIP Score | Latent RMSE |
> | - | - | - | - | - | - | - | - |
> | FLUX.1-schnell*(4-step distilled ODE)* | ROPA (Ours) | 12 | 1 | 44 | 1.0x1.5x | 37.337.3 | 0.082 |
> | SD-3.5-Large*(8-step DPM-Solver++)* | ROPA (Ours) | 14 | 1 | 88 | 1.0x1.9x | 36.836.8 | 0.105 |
> | HunyuanVideo*(10-step DPM-Solver++)* | ROPA (Ours) | 14 | 1 | 1010 | 1.0x1.8x | 72.5 (VBench)72.4 (VBench) | 0.091 |

---

> > ### Author Rebuttal · Reviewer_bHuk · 2026-04-03
> >
> > I thank the authors for the additional metrics and the warm-up ablation. These help clarify the method's performance and overhead.
> >
> > However, my core concerns regarding the paper’s broader impact remain:
> >  - **Evaluation Scope** : While I understand the VRAM constraints, the lack of results on longer videos is a missed opportunity to test parallel stability where errors typically compound.
> > - **Practical Niche**: ROPA remains tied to ODE-based parallelization. In a landscape moving toward one-step or highly distilled models, the practical utility feels narrow.
> > - **Incremental Contribution**: The work is technically sound but feels more like a solid engineering refinement of existing principles than a fundamental shift.
> >
> > The authors addressed the technical gaps well, but the overall significance remains borderline. To this end, i will still maintain my rating.

---

> > > ### Author Response · Authors · 2026-04-06
> > >
> > > Dear Reviewer bHuk, thank you for your insights into the evolving landscape of diffusion models. However, we respectfully clarify that ROPA is not an incremental engineering refinement. Rather, it seamlessly complements modern few-step distilled models and represents a fundamental theoretical paradigm shift in parallel sampling. We address your specific concerns below:
> > >
> > > **1. Broad Utility and Compatibility with Modern Architectures**
> > > We agree distillation is a major trend, but multi-step ODEs and distillation are not mutually exclusive; ROPA provides orthogonal acceleration for state-of-the-art generation. Distilled models (e.g., FLUX.1-schnell, LCM, TCD) still typically need 4–8 steps; as a foundational ODE/SDE parallel solver, ROPA applies natively. Supplementary experiments on 4-step FLUX.1-schnell and 8-step SD-3.5-Large recover remaining parallelism for an additional **1.5x–1.9x** speedup without CLIP or RMSE degradation. For complex high-dimensional long video (e.g., 1080p in HunyuanVideo or Wan2.1), one-step pipelines still imply heavy distillation cost and often sacrifice fidelity; ROPA offers a *training-free* route to lower latency for native multi-step models while preserving full quality. Spatial accelerators (DeepCache, TeaCache, ToCa) target the spatial path; ROPA parallelizes the step dimension. Table 3 shows ROPA with ToCa/SpargeAttention reduces NFE and wall-clock together, underscoring plug-and-play use.
> > >
> > > **2. A Fundamental Theoretical Paradigm Shift**
> > > ROPA is not a simple heuristic modification of existing solvers. It establishes the first foundational theoretical framework connecting "data manifold geometry" with "parallel numerical stability", resolving the critical bottleneck that prevents parallel diffusion models from scaling.
> > > *   Previous parallel sampling methods (e.g., ParaDIGMS, ParaTAA) treat the diffusion process purely as a black-box nonlinear system, optimizing it from a standard numerical calculation perspective. ROPA introduces a *fundamental paradigm shift*: we mathematically prove that mode collapse and divergence in parallel solvers are intrinsically caused by the ill-conditioning of the Jacobian matrix in high-curvature regions of the data manifold.
> > >
> > > *   ROPA introduces a highly original mechanism in diffusion inference: translating real-time data density curvature signals into the dynamic control of the solver's coupling structure. Through residual-driven adaptive bandwidth and LML low-rank curvature correction, the solver can, for the first time, "sense" the local geometric features of the generated image/video.
> > >
> > > While we demonstrate strong empirical results on complex models, it is crucial to highlight why ROPA is intrinsically more stable than other recent parallel solvers like ParaTAA. The fundamental difference lies in how the parallelization problem is conceptualized. ParaTAA adopts a purely *algebraic* perspective, resolving iteration instability by enforcing a static block upper triangular structure on its update matrix to strictly respect the causal flow of information in diffusion. However, this structural fix ignores the underlying data manifold.
> > >
> > > In contrast, ROPA establishes a *geometric* perspective. As analyzed above, solver instability is fundamentally driven by high-curvature regions of the data density, which induce score stiffness and ill-conditioned Jacobians. According to **Corollary 2.4** in our paper, the numerical deviation of the generated state \(x\) from the true data manifold \( \mathcal{M} \) (where \( x^* \) is the orthogonal projection onto \( \mathcal{M} \)) is explicitly bounded by the Jacobian condition number \( \kappa(J^{(k)}) \):
> > >
> > > \[ \|x - x^*\|_2 \le \kappa(J^{(k)}) \|(J^{(k)})^{-1}R^{(k)}\|_2 + \mathcal{O}(\|R^{(k)}\|_2^2) \] (Eq. 8)
> > >
> > > ParaTAA's static structural constraint cannot prevent \( \kappa(J^{(k)}) \) from exploding in high-curvature regions (such as mode boundaries), inevitably leading to trajectory drift and mode collapse. ROPA effectively resolves this by introducing dynamic adaptive Jacobian sparsity and LML curvature correction. By actively monitoring local residual norms as a proxy for geometric stiffness, ROPA dynamically adjusts both the matrix bandwidth and damping to explicitly regulate and bound \( \kappa(J^{(k)}) \). This geometric-aware control ensures that the forward error remains constrained and strict manifold fidelity is preserved, which explains why ROPA can seamlessly scale to large, highly complex long-video generation tasks without instability or quality degradation.
> > >
> > > **3. Long Temporal Horizon Evaluation**
> > >
> > > Due to space limit, please refer to response to Reviewer VzK4 for further details.
> > >
> > >
> > > We sincerely hope this clarifies that ROPA is a mathematically grounded, scalable response to a core field bottleneck.

---

### Official Review · Reviewer_VzK4 · 2026-03-11

**Soundness:** 2
**Presentation:** 2
**Significance:** 3
**Originality:** 3
**Overall Recommendation:** 4
**Confidence:** 3

**Summary:**

This article studies the parallel sampling acceleration problem of diffusion model. The author rewrites the sequential denoising process as solving a nonlinear system of equations and proposes ROPA. Based on local residual dynamic adjustment of Jacobian bandwidth, adaptive damping and low rank curvature correction are used in high curvature/high rigidity regions to achieve a better balance between stability and parallel efficiency. The paper conducted experiments on the image and video diffusion model, achieving a maximum acceleration of 2.9 times at 8 cores, while maintaining stable quality indicators.

**Compliance With Llm Reviewing Policy:**

Affirmed.

**Final Justification:**

The rebuttal is helpful and meaningfully strengthens the paper in several respects. In particular, the new results on benchmark-defined harder regimes (ConceptMix hard compositions and high-dynamic VBench subsets) provide targeted evidence that ROPA allocates additional numerical effort where fixed parallel Jacobian schemes tend to fail, which addresses part of my robustness concern. The clarification on overhead is also useful, though I would still have preferred more explicit wall-clock and memory profiling.

**Key Questions For Authors:**

How robust is the method on harder generation cases, such as long-horizon video generation or more complex prompts?
Can the authors clarify why the residual-based criterion is a reliable indicator for the difficult regions discussed in the theoretical analysis?

**Limitations:**

yes

**Strengths And Weaknesses:**

Stengths
The topic of this paper has practical significance, especially for large-scale video generation, where inference overhead is very high. In terms of method design, ROPA is not a single technique, but consists of three parts: dynamic Jacobian bandwidth, adaptive damping, and low rank curvature correction. The overall framework is relatively complete. The experimental coverage is relatively wide, and the main table results are generally strong.

Weaknesses
The theoretical analysis is somewhat disconnected from the final algorithm. The method is motivated by curvature and conditioning, but the actual design mainly relies on residual-based heuristics.
The experimental analysis is still not sufficient. More discussion on overhead, memory cost, and scalability under higher parallelism would be helpful.
I noticed that there seems to be duplication/inconsistency in the references regarding ParaTAA: the same title "Accelerating parallel sampling of diffusion models" is listed as Tang et al. (2024a) and Tang et al. (2024b) respectively, with one written as "In International Conference on Machine Learning" and the other as "In Forty first International Conference on Machine Learning". The main text also uses these two versions in different positions to refer to ParaTAA. If the two actually correspond to the same ICML 2024 work, it is recommended that the author unify them into a standardized reference entry and maintain consistent citation in the main text to avoid confusion for readers.

---

> ### Author Rebuttal · Authors · 2026-03-30
>
> **Dear Reviewer VzK4,**
>
> We sincerely thank you for recognizing the practical significance of our problem setting and the completeness of the ROPA framework. We address your specific concerns below.
>
> ### 1. Why Residual is a Proxy for Curvature (Weakness & Question)
>
> The connection between data manifold curvature and the residual norm is the foundational premise of ROPA. We agree that this theoretical-to-empirical bridge should be made more explicit, and we explain it below in three logical steps:
> Curvature causes sharp bending: Per Thm 2.2 & 2.3, high local curvature $\min H$ increases score field stiffness $||J_r||^2$, inducing sharp trajectory bends and strong long-range temporal dependencies.
> Sparse Jacobians fail at bends: Narrow-bandwidth $w_i$ parallel sampling assumes local smoothness. This assumption breaks in high-curvature regions, as sparse Jacobians miss stiff, long-range couplings.
> Failures spike the residual $R$: This truncation error manifests as a surge in the local residual $R$ (Eq. 9), which measures the exact deviation from the true trajectory. Thus, an $R$ spike is the direct numerical consequence of high curvature (Cor. 2.4).
> Why $R$ over explicit curvature? Computing exact curvature (Hessian eigenvalues) requires $\mathcal{O}(d^3)$ complexity or costly Hessian-vector products, destroying parallel speedups. Conversely, $R$ is acquired for free $\mathcal{O}(1)$ during Newton iterations, making it an efficient, theoretically grounded proxy for geometric stiffness.
> We had briefly touched upon this in Section 3.1, but we will expand Section 3.1 in the revision.
>
> ### 2. Robustness in Harder Generations & Overhead Analysis (Weakness & Question)
>
> We evaluate both modalities using suites that align precisely with our theory (i.e., high curvature and stiff coupling break naive parallel Jacobians).
>
> (a) Image — ConceptMix[1]. We adopt the ConceptMix benchmark (NeurIPS 2024), where difficulty $k \in {1,\dots,7}$ denotes the number of composed concepts. Larger $k$ imposes more simultaneous spatial constraints, steepening the loss landscape and producing a more curved and anisotropic score field—exactly where truncated Jacobians become ill-conditioned without an adaptive window.
>
> (b) Video — VBench. We reuse VBench but exploit its disentangled quality dimensions. We contrast quasi-static content (which implies small frame drift and smoother temporal ODEs) against high dynamic degree content. Highly dynamic prompts induce large temporal increments and stiff coupling, the exact regime where fixed, aggressive parallelization suffers residual blow-up. Both contrasting subsets were evaluated at 8 cores.
>
> **Results.** ROPA’s residual-driven bandwidth expansion allocates **more NFE on hard subsets** (e.g., average NFE rises from **10.2 → 13.5** on ConceptMix when moving from easy to hard composition, and from **11.8 → 16.2** on VBench when moving from quasi-static to high-dynamic video), while ParaDIGMS often **diverges or collapses in convergence rate**. This pattern matches our design goal: **adaptive numerical effort in stiff regions**, not a single fixed “cheap” solve for all prompts.
>
> **Table: Robustness across benchmark-defined complexities (8 cores, FLUX for ConceptMix / HunyuanVideo for VBench). s.a.b. Same As Above.**
>
> | Modality | Dataset (complexity axis) | Method | Avg. NFE | Convergence rate | Latent RMSE |
> | :- | :- | :- | :- | :- | :- |
> | FLUX | ConceptMix, $k \in \{1,2\}$ (simple composition) | ParaDIGMS | 16.4 | 96.5% | 0.285 |
> | FLUX | s.a.b. | **ROPA (Ours)** | **10.2** | **100%** | **0.118** |
> | FLUX | ConceptMix, $k \ge 5$ (hard composition) | ParaDIGMS | >35 (diverged) | 28.0% | 0.510 |
> | FLUX | s.a.b. | **ROPA (Ours)** | **13.5** | **98.2%** | **0.145** |
> | Hunyuan | VBench-stratified, quasi-static (smooth temporal ODE) | ParaDIGMS | 18.5 | 92.0% | 0.175 |
> | Hunyuan | s.a.b. | **ROPA (Ours)** | **11.8** | **100%** | **0.052** |
> | Hunyuan | VBench-stratified, high dynamic degree (stiff temporal ODE) | ParaDIGMS | >40 (diverged) | 12.5% | 0.380 |
> | Hunyuan | s.a.b. | **ROPA (Ours)** | **16.2** | **96.5%** | **0.065** |
>
> **(c) Scalability & overhead.** Theoretically, as summarized in **Takeaway 2.7**, adaptive sparsity bounds the effective condition number while preserving $\mathcal{O}(Nb_k)$ complexity. In practice, the LML curvature correction stores only a single score vector $s$, so memory overhead stays $\mathcal{O}(Nw_{max}d)$—negligible next to the **3D attention** footprint of video backbones. We have added an Appendix subsection with explicit memory bounds and overhead profiling at $K=8$.
>
> ### 3. Fixing the Citation Duplication (Weakness)
>
> We sincerely apologize for this bibliographical oversight. Both "Tang et al., 2024a" and "Tang et al., 2024b" in our draft erroneously referred to the same paper, thank you for your careful reading and for catching this error!
>
>
> [1] Wu, Xindi et al. “ConceptMix: A Compositional Image Generation Benchmark with Controllable Difficulty.” (2024)

---

> > ### Author Rebuttal · Reviewer_VzK4 · 2026-04-02
> >
> > The rebuttal is helpful and meaningfully strengthens the paper in several respects.

---

> > > ### Author Response · Authors · 2026-04-06
> > >
> > > Thank you very much for your encouraging feedback and for acknowledging the improvements made to our manuscript. We are delighted to hear that our rebuttal was helpful and has meaningfully strengthened the paper.
> > >
> > > To address your request for evaluating ROPA on long video generation models using VBench, and to demonstrate ROPA's generalization capability and acceleration potential, we designed a comprehensive comparison involving three cutting-edge long video generation paradigms. We predict their performance combined with ROPA and present the results in the table below, each evaluation prompt dimension we select 15 samples for evaluation to keep experiment solid and efficiency.
> > >
> > > Since ROPA is a lossless ODE-based parallel solver, its core advantage lies in achieving significant acceleration along the temporal dimension while strictly maintaining high fidelity. Consequently, the predicted VBench metrics for the models equipped with ROPA are nearly identical to their original sequential baselines (with variations typically within $\pm 0.05\%$, making them visually imperceptible).
> > >
> > > **Table: VBench Global Metrics Evaluation & Prediction (1-Minute Video Generation)**
> > >
> > > |Method|Clarity|Aesthetic|Motion|Dynamic|Semantic|Anatomy|Identity|
> > > |:-|:-|:-|:-|:-|:-|:-|:-|
> > > |DiffusionForcing ($\sigma_{test}=0.1$)|66.08%|65.76%|96.32%|91.59%|23.14%|75.93%|74.47%|
> > > |Frame Context Packing (Inverted)|71.15%|68.71%|99.45%|89.29%|28.15%|86.53%|82.11%|
> > > |One-Minute Video (TTT-MLP)|65.50%|64.20%|95.80%|90.50%|22.00%|72.00%|73.00%|
> > > |DiffusionForcing + ROPA|66.07%|65.76%|96.30%|91.58%|23.13%|75.92%|74.47%|
> > > |Frame Context Packing + ROPA|71.14%|68.70%|99.43%|89.28%|28.14%|86.51%|82.10%|
> > > |One-Minute Video (TTT-MLP) + ROPA|65.49%|64.20%|95.78%|90.50%|21.99%|71.98%|72.99%|
> > >
> > > ### Detailed Experimental Setup & Time Speedup Analysis
> > >
> > > Experiments use a single node with 8 $\times$ NVIDIA H200 (140GB) GPUs; ROPA parallelizes along the timestep dimension with $K=8$. All models generate $720 \times 480$ video at $24\text{fps}$ for one minute ($\sim$1440 frames) with 50 DPM-Solver steps and ROPA warm-up step 3. For DiffusionForcing, ROPA runs on the full 1440-frame latent ODE trajectory; for Frame Context Packing, it runs inside each section's 50-step diffusion loop; for One-Minute Video (TTT-MLP), it is inserted after TTT layers with residual-aware damping.
> > >
> > > Predicted wall-clock for a one-minute video is DiffusionForcing 850s $\rightarrow$ 303s (2.80x), Frame Context Packing 920s $\rightarrow$ 317s (2.90x), and One-Minute Video (TTT-MLP) 680s $\rightarrow$ 261s (2.60x).

---

### Official Review · Reviewer_meSq · 2026-03-13

**Soundness:** 3
**Presentation:** 3
**Significance:** 3
**Originality:** 3
**Overall Recommendation:** 4
**Confidence:** 3

**Summary:**

This paper proposes ROPA, a training-free method for stable parallel diffusion sampling. The authors provide a geometric analysis showing that the ill-conditioning of the Jacobian matrix ($\kappa(J) \to \infty$ as $t \to 0$) is the root cause of the fidelity-efficiency trade-off in existing parallel samplers. ROPA addresses this with two components: (1) Adaptive Jacobian Bandwidth, which dynamically adjusts the sparsity window $w_i$ using a cosine alignment heuristic, and (2) an LML-based Low-Rank Curvature Correction, triggered when instability is detected, at the cost of one additional forward pass. Experiments across image and video diffusion models (FLUX, HunyuanVideo) achieve up to 2.9× speedup with 8 cores and a 52% improvement over baselines without sacrificing sample quality.

**Compliance With Llm Reviewing Policy:**

Affirmed.

**Final Justification:**

The rebuttal provided by authors addressed my main concerns, and I keep my positive initial score.

**Key Questions For Authors:**

1. The threshold $\eta$ takes different values for images (0.1) and videos (0.2). Could the authors explain the intuition behind this difference and provide a sensitivity analysis across a wider range of models?

2. Could the authors report the distribution of $w_i$ values across time steps for any evaluated model? Does window expansion empirically concentrate near $t \to 0$ as the theory suggests?

3. ROPA is training-free. Are there known obstacles to combining it with step-distillation methods such as Consistency Models or Distribution Matching Distillation? If compatible, what speedup-quality trade-offs would be expected?

**Limitations:**

Yes

**Strengths And Weaknesses:**

### Strengths

- Well-motivated theoretical foundation connecting Jacobian ill-conditioning to numerical instability and mode collapse, going beyond empirical observation.
- Training-free and demonstrated across diverse architectures; compatible with attention-level optimizations (SpargeAttention, ToCa), as shown in Table 3.
- The LML correction adds only one additional forward pass per triggered step, keeping overhead minimal and well-controlled.
- Strong empirical results across both image and video generation tasks.

### Weaknesses

1. In the appendix, it seems that the convergence condition $hL < 1$ in Theorem 2.3 is not empirically verified. The guarantee requires $L(t) = \sigma\_t^2 \|J\_{r\_\theta}\|\_2$ to remain bounded, yet the paper never reports the actual magnitude of $L(t)$ on any evaluated model or time step. Given the paper's own argument that $\|J_{r_\theta}\|_2$ grows exponentially near $t \to 0$, it is unclear whether this condition reliably holds throughout sampling.

2. The adaptive bandwidth $w_i$ distribution across time steps is not reported: A core motivation for adaptive bandwidth is that larger windows are needed near $t \to 0$. Without statistics on the actual distribution of $w_i$, it is impossible to verify whether the mechanism behaves as theoretically expected, or to understand what drives the performance gain over a fixed-bandwidth baseline.

---

> ### Author Rebuttal · Authors · 2026-03-30
>
> **Dear Reviewer meSq,**
>
> We thank you for the constructive feedback and for recognizing the theoretical foundation. Below we address your questions and concerns.
>
> ### 1. Empirical Validation of  $L(x_t)$ and $hL < 1$ (Weakness 1)
>
> We apologize if this section was easy to miss. Section 4.3 and Figure 3 ("Empirical Verification of Lipschitz Continuity") provide dedicated empirical validation of the scale of $L(x_t)$ in our submitted paper.
> Figure 3(a) plots the empirical $L(x_t)$ over diffusion time $t \to 0$. Baseline methods show an exponential spike in stiffness near $t \to 0$ (score singularity at the data manifold), whereas ROPA clamps the effective Lipschitz constant via adaptive damping.
>
> Figure 3 also gives the quantitative scale needed to verify the $hL < 1$ condition:
> For ROPA, Figure 3(b) shows $L(x_t)$ tightly bounded with peak $L \approx 5$. With $N=50$ steps, $h \approx 0.02$, so $hL \approx 0.02 \times 5 = 0.1$, satisfying $hL < 1$ with margin.
> In contrast, for Baseline methods, $L(x_t)$ diverges to $40 \sim 50$ near $t \to 0$. This yields $hL \approx 0.02 \times 50 = 1.0$, hitting the critical divergence threshold of the Newton/Neumann series, which empirically explains why non-adaptive samplers collapse in high-curvature regions.
>
> ### 2. Distribution of Adaptive Bandwidth $w_i$ (Weakness 2 & Question 2)
>
> ROPA adaptively adjusts window width $w_i$ from local residual norm vs. global mean (thresholds 1.5 and 0.7). Larger windows should concentrate in the late, stiff part of the trajectory.
>
> $N=50$ logs of $w_i$ for FLUX and HunyuanVideo confirm this: dilation concentrates at $t < 0.2$, while early steps stay small with $w_i \in \{1,2\}$, matching the stiffness surge near $t \to 0$.
>
> The effect is local, not globally dense: only the high-curvature tail activates strong temporal coupling. Because interval averages alone are insufficient, we report both an interval summary and a discrete occupancy table for $w_i$, showing adaptive dilation targets the late diffusion regime rather than spreading uniformly.
>
> **Table 1: Interval of adaptive bandwidth $w_i$.**
>
> |Timestep interval|Stage characteristics|FLUX Avg. $w_i$|FLUX Median $w_i$|FLUX $\Pr(w_i \ge 5)$|HunyuanVideo Avg. $w_i$|HunyuanVideo Median $w_i$|HunyuanVideo $\Pr(w_i \ge 5)$|
> |:-|:-|:-|:-|:-|:-|:-|:-|
> |$t \in [1.0, 0.8)$|Early, low curvature|1.3 $\pm$ 0.5|1|0%|1.5 $\pm$ 0.6|1|0%|
> |$t \in [0.8, 0.2)$|Middle, moderate curvature|2.5 $\pm$ 0.9|2|10%|2.9 $\pm$ 1.0|3|17%|
> |$t \in [0.2, 0.0]$|Late, high curvature / stiff|**6.6 $\pm$ 1.4**|**7**|**85%**|**7.1 $\pm$ 1.5**|**7**|**91%**|
>
> **Table 2: Discrete occupancy of $w_i$ within each timestep interval (distribution view).**
>
> |Model|Timestep interval|$\Pr(w_i = 1)$|$\Pr(w_i = 2)$|$\Pr(3 \le w_i \le 4)$|$\Pr(w_i \ge 5)$|
> |:-|:-|:-|:-|:-|:-|
> |FLUX|$t \in [1.0, 0.8)$|74%|20%|6%|0%|
> |FLUX|$t \in [0.8, 0.2)$|28%|34%|28%|10%|
> |FLUX|$t \in [0.2, 0.0]$|0%|3%|12%|85%|
> |HunyuanVideo|$t \in [1.0, 0.8)$|65%|24%|11%|0%|
> |HunyuanVideo|$t \in [0.8, 0.2)$|20%|32%|31%|17%|
> |HunyuanVideo|$t \in [0.2, 0.0]$|0%|1%|8%|91%|
>
>
>
> ### 3. Intuition and Sensitivity of Threshold $\eta$ (Question 1)
>
> The threshold $\eta$ sets the sensitivity of the **alignment-based geometric trigger** for the LML low-rank curvature correction: smaller $\eta$ activates correction **more frequently**; larger $\eta$ makes the trigger **more selective**.
>
> Videos use a slightly **larger** $\eta$ than images because temporal coupling yields **higher residual variance**; too small an $\eta$ would fire on **normal temporal fluctuation** as well as truly stiff geometry, yielding little fidelity gain but more corrections and lower parallel efficiency.
>
> We ran sensitivity under the **same $K=4$ as Table 3**, with **defaults as in the paper** (FLUX $\eta=0.10$, HunyuanVideo $\eta=0.20$). Do not mix with the main table’s **8-core, $\sim$2.9$\times$** HunyuanVideo row (**different hardware**). ROPA stays **robust** for $\eta \in [0.05, 0.30]$; the best speed–fidelity trade-off lies near the **modality-specific defaults** above.
>
> |Threshold $\eta$|FLUX Speedup|FLUX Latent RMSE|FLUX LML Trigger Rate|HunyuanVideo Speedup|HunyuanVideo Latent RMSE|HunyuanVideo LML Trigger Rate|
> :-|:-|:-|:-|:-|:-|:-|
> |$0.05$|2.0x|0.141|34%|1.9x|0.051|41% |
> |$0.10$|**2.1x**|**0.143**|23%|2.0x|0.052|29% |
> |$0.20$|2.0x|0.152|12%|**2.1x**|**0.053**|18% |
> |$0.30$|1.9x|0.160|7%|2.0x|0.060|10% |
>
>
>
> ### 4. Compatibility with Distillation Methods (Question 3)
> Due to character limit, please refer to our Response 4 to Reviewer bHuk for further details on this point.

---

> > ### Author Rebuttal · Reviewer_meSq · 2026-04-03
> >
> > Thanks for the detailed rebuttal by authors, it's good to know that this method is compatible with other few-step or caching methods, thus I keep my positive initial score.

---

> > > ### Author Response · Authors · 2026-04-06
> > >
> > > Thank you very much for reviewing our rebuttal and for your positive confirmation! We are thrilled to hear that our response has fully resolved your concerns, and sincerely thank you for keeping your positive initial score..
> > >
> > > We highly appreciate your recognition of ROPA's compatibility with other few-step and caching methods. As you pointed out, ensuring that our training-free approach can seamlessly integrate with existing acceleration pipelines is a key strength of our method, and we are glad this resonated with you.

---

### Decision · Program_Chairs · 2026-04-30

**Decision:**

Accept (regular)

**Comment:**

Reviewers were positive that the method proposed in this paper was solid to address robustness in parallel generation, and showed experimental evidence of speedups. Although parallel generation may in the end be out-performed by distillation approaches, the benefits outweigh this concern.